# Vision-Language Models Create Cross-Modal Task Representations

**Grace Luo** [1]  **Trevor Darrell** [1]  **Amir Bar** [1]

`vlm-cross-modal-reps.github.io`

## Abstract

Autoregressive vision-language models (VLMs) can handle many tasks within a single model, yet the representations that enable this capability remain opaque. We find that VLMs align conceptually equivalent inputs into a shared task vector, which is invariant to modality (text, image) and format (examples, instruction), and may simplify VLM processing. We measure this alignment via cross-modal transfer–the ability of a task vector derived in one modality to trigger the correct generation in another–on a range of tasks and model architectures. Although the task vector is highly compressed, we find that this single vector outperforms prompting the model with the full task information, unique to this cross-modal case. Furthermore, we show that task vectors can be transferred from a base language model to its fine-tuned vision-language counterpart, and that they can be derived solely from instructions without the need for examples. Taken together, our findings shed light on how VLMs internally process task information, and how they map different modalities into common semantic representations.

## 1. Introduction

Depending on the input instruction (Liu et al., 2023a) or in-context examples (Alayrac et al., 2022), VLMs can dynamically adjust the task executed on the image. This flexibility poses a challenge: each task can be defined in a different way, and memorizing every possible variation is impractical. There needs to exist some form of compression, or representation sharing, to manage this complexity.

This brings into question whether the many ways of describing the same underlying task converge to a shared VLM

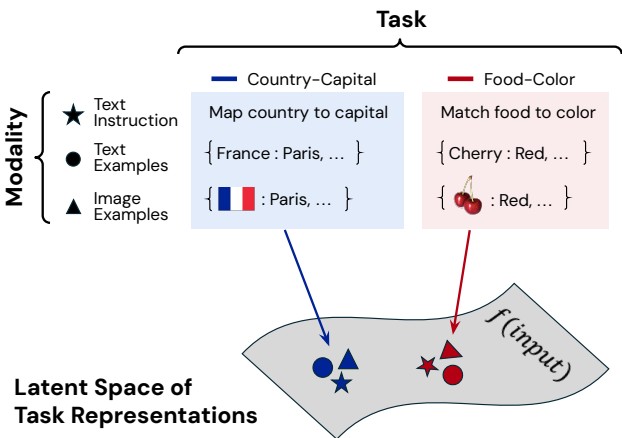

Figure 1: VLMs map conceptually equivalent inputs into a shared task representation. This representation is invariant to the specification, regardless of modality (image, text) and format (examples, instruction).

representation. Consider the tasks shown in Figure 1, which can be equivalently expressed using instructions, text examples, or image examples. One might argue that the model is biased towards learning the same simple function that solves the task, for all specifications (Solomonoff, 1964; Valle-Perez et al., 2019; Lin et al., 2023; Huh et al., 2024).

In this work, we provide evidence of such a shared task representation, which is invariant to specification modality or format. This representation exists at a special token position near the end of the sequence, which is contextualized by the inputs to form a high-level summary, also known as the task vector (Hendel et al., 2023; Todd et al., 2024). This observation is non-trivial; recall that VLM training is designed to encourage the alignment of image and text embeddings, but not the emergent task summaries derived from them. This difference is apparent from the t-SNE (van der Maaten & Hinton, 2008) visualization in Figure 2. While the image and text embeddings exhibit no task-specific grouping (a), the task vectors cluster by the color-coded tasks (b).

To quantify the alignment of these task representations, we evaluate cross-modal transfer. For example, we evaluate the VLM's ability to apply a task expressed with text examples onto an image query (see Figure 3). We find that "patch-

---

[1]University of California, Berkeley, USA. Correspondence to: Grace Luo <graceluo@berkeley.edu>.

*Proceedings of the 42nd International Conference on Machine Learning*, Vancouver, Canada. PMLR 267, 2025. Copyright 2025 by the author(s).

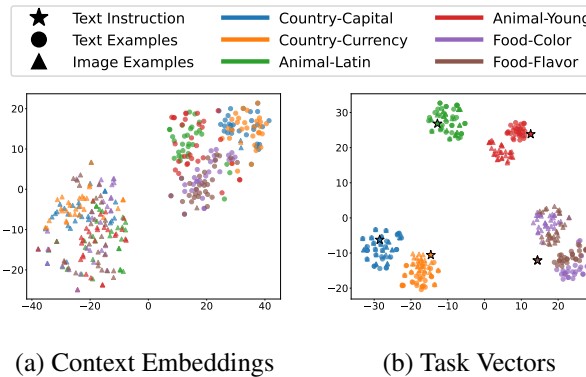

(a) Context Embeddings    (b) Task Vectors

Figure 2: **t-SNE clustering.** The representation at a final token position, the task vector, summarizes the preceding context. At the same intermediate layer, unlike (a) the image and text embeddings, (b) the task vectors cluster by task (color) not modality (shape).

ing" (Zhang & Nanda, 2023) the task vector, or injecting the representation without otherwise specifying the task, often induces the model to generate the correct answer.

Here, we summarize our most intriguing findings. First, although patching utilizes only a single vector, it significantly outperforms few-shot prompting (Brown et al., 2020) with the uncompressed cross-modal examples. Second, while one might expect VLM fine-tuning to significantly alter representations, we show that task vectors transfer between the base LLM and its corresponding VLM. Third, we show that task vectors can be defined with instructions, an unexplored alternative that is also complementary with examples. Finally, we analyze the answer generation process, and we find that VLMs map different specifications into similar task vectors but maintain distinct image and text embeddings.

## 2. Related Work

**Mechanistic Interpretability.** The goal of mechanistic interpretability is to make deep models more transparent by interpreting how and why model decisions are made (Gilpin et al., 2018; Gurnee & Tegmark; Liu et al., 2022; Geva et al., 2020; Nanda et al., 2023). To uncover the relationships within the model, *causal interventions* (Pearl, 2022) are often used. For example, activation patching (Zhang & Nanda, 2023) is a technique used to modify neural network activations to observe changes in outputs, often with causal insights to correct biased or erroneous behavior (Meng et al., 2022; Bau et al.). We use activation patching to demonstrate that task representations transfer across modalities.

**In-Context Learning.** With the advent of large language models (LLMs) (Brown et al., 2020), researchers have sought to explain in-context learning (Liu et al., 2023b),

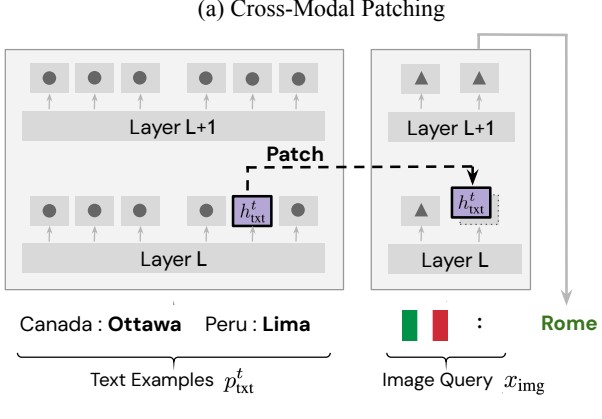

(a) Cross-Modal Patching

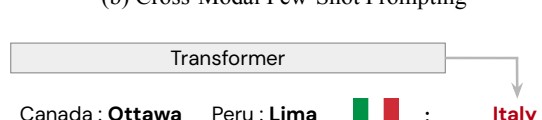

(b) Cross-Modal Few-Shot Prompting

Figure 3: **Cross-modal transfer**. (a) A single compressed task vector from one modality can induce the VLM to perform the task on queries from another modality, without additional training; this outperforms (b) feeding the full task information (see Table 2).

the phenomenon where LLMs can adapt to new tasks with a few input examples in the forward pass. Olsson et al. (2022) hypothesized that ICL is driven by attention heads ("induction heads"), while Xie et al. (2021) interprets ICL as an implicit Bayesian Inference process, and Garg et al. (2022) showed that ICL can emerge in the simple case of linear functions. More recently, Hendel et al. (2023), Todd et al. (2024), Liu et al. (2024b) hypothesized that ICL creates task (or function) vectors, latent activations that encode the task in LLMs, and Hojel et al. (2024) demonstrated a similar behavior in computer vision models. Huang et al. (2024) proposed to use task vectors in VLMs to compress long prompts that would otherwise not fit in a limited context length. We uniquely study the cross-modal properties of VLM task representations, and how conceptually equivalent specifications can lead to similar representations.

**Vision-Language Models.** Recent VLMs can be categorized into modality late-fusion (Liu et al., 2023a; 2024a; Li et al., 2023; Tong et al., 2024) and early-fusion (Bavishi et al., 2023; Lu et al., 2022; 2023; Team, 2024; Zhou et al., 2024) approaches. Late-fusion approaches typically combine a pre-trained visual encoder and LLM by training adapters, potentially with a short end-to-end fine-tuning stage. In contrast, early-fusion approaches focus on end-to-end training without any pre-initialization of the representations. We observe cross-modal task representations in both categories, suggesting that this property can emerge regard-

Table 1: **Cross-modal tasks.** We design six tasks inspired by the text examples in prior work (Hendel et al., 2023; Todd et al., 2024), where we add alternative specifications such as instructions and image examples.

| Task | Instruction | Text Example | Image Example |
|---|---|---|---|
| Country-Capital | *The capital city of the country:* | {Greece : **Athens**} | {  : **Athens**} |
| Country-Currency | *The last word of the official currency of the country:* | {Italy : **Euro**} | {  : **Euro**} |
| Animal-Latin | *The scientific name of the animal's species in latin:* | {Gray Wolf : **Canis lupus**} | {  : **Canis lupus**} |
| Animal-Young | *The term for the baby of the animal:* | {Common Dolphin : **calf**} | {  : **calf**} |
| Food-Color | *The color of the food:* | {Persimmon : **orange**} | {  : **orange**} |
| Food-Flavor | *The flavor descriptor of the food:* | {Strawberry : **sweet**} | {  : **sweet**} |

less of the initialization. Several works examine image ICL in VLMs, where they propose new models (Alayrac et al., 2022; Laurençon et al., 2024; Jiang et al., 2024) and analyze the impact of example selection on performance (Doveh et al., 2024; Baldassini et al., 2024). Our work offers a new perspective on image ICL by comparing it with text ICL and demonstrating the similarity between the two processes. We even show that LLaVA (Liu et al., 2023a), which lacks image ICL capabilities, can still benefit from text ICL.

## 3. Cross-Modal Task Representations

In this work, we are interested in studying the task representations of VLMs. We assess how *cross-modal* these representations are, or the extent to which the model aligns inputs from different modalities into shared task representations. We outline common VLM task specifications in Sec. 3.1, followed by our cross-modal patching formulation, where we show a task vector can be derived from one modality and transferred to another, in Sec. 3.2.

### 3.1. Task Specifications for VLMs

To study this question, we start by designing an evaluation set of cross-modal tasks. This evaluation set should contain different specifications that refer to the same underlying task, so that we can evaluate the alignment of their task representations.

**Specifications.** Below, we enumerate three common task specifications for prompting VLMs, which we study in our work (also see Table 1). Given that VLMs are primarily used for image analysis, in this work we focus on image queries and discuss each setting with this in mind.

1. **Text Examples.** We first measure *cross-modal transfer*. We investigate whether functions defined via text examples can generalize to image queries in Sec. 4.1. We also

look at a special case of transferring task information from a base LLM to its fine-tuned VLM in Sec. 4.2.
2. **Instructions.** Beyond the examples studied in prior work, we also consider instructions in Sec. 4.3. We also evaluate *ensembling* the instructions and examples to better convey the task. Finally, in Sec. 4.5 we explore task *overriding*, where the goal is to override a task locally defined in the prompt with a global instruction.
3. **Image Examples.** For image queries, image examples represent the unimodal baseline, while text examples and instructions are considered cross-modal. We compare how the model processes image versus text examples by analyzing *representation evolution* in Sec. 5. Similarities between the processes suggest cross-modal specifications are a valid alternative to unimodal ones.

**Evaluation Tasks.** We construct six tasks, each defined by a single text instruction, a pool of text examples, or a pool of image examples, as seen in Table 1. For each task, we split the example pool into 30 samples for validation and 100 for testing, where the split is kept consistent across modalities. Each sample is then used as a query, where its corresponding answer is the ground-truth label. To create an example-based specification, we randomly select $N$ samples without replacement, while ensuring no overlap with the query. We provide more details in Sec. A.1 of the Appendix.

### 3.2. Cross-Modal Patching

Although we observe initial evidence of shared representations via clustering (Figure 2), we need a more rigorous method for quantifying alignment. To this end, we propose *cross-modal patching*, where we transfer the representation across modalities and measure the effect on model outputs. We observe that a task representation derived in one modality can be used to trigger the correct generation in another. We discuss our method in detail next.

Table 2: **Cross-modal transfer results**. We display the accuracy across six tasks on an unseen test set. For image queries, patching cross-modal task vectors (Text Examples Patch) outperforms few-shot prompting (Text Examples Prompt) and the strong unimodal baselines (Image Examples Prompt, Patch). The best method per task is underlined and overall is **bolded**. We also denote whether the method is cross-modal (✓) or not (×).

| Method | Cross-Modal? | Country-Capital | Country-Currency | Animal-Latin | Animal-Young | Food-Color | Food-Flavor | Avg. |
|---|---|---|---|---|---|---|---|---|
| Random | - | 0.00 | 0.12 | 0.00 | 0.18 | 0.24 | 0.31 | 0.14 |
| **LLaVA-v1.5** | | | | | | | | |
| No Context | - | 0.00 | 0.00 | 0.00 | 0.00 | 0.00 | 0.00 | 0.00 |
| Image Examples Prompt | × | - | - | - | - | - | - | - |
| Image Examples Patch | × | - | - | - | - | - | - | - |
| Text Examples Prompt | ✓ | 0.02 | 0.18 | 0.03 | 0.23 | 0.28 | 0.37 | 0.18 |
| Text Examples Patch | ✓ | 0.31 | 0.30 | 0.26 | 0.18 | 0.53 | 0.31 | **0.32** |
| **Mantis-Fuyu** | | | | | | | | |
| No Context | - | 0.00 | 0.00 | 0.00 | 0.00 | 0.00 | 0.00 | 0.00 |
| Image Examples Prompt | × | 0.11 | 0.13 | 0.24 | 0.05 | 0.34 | 0.23 | 0.18 |
| Image Examples Patch | × | 0.17 | 0.03 | 0.16 | 0.05 | 0.50 | 0.31 | 0.20 |
| Text Examples Prompt | ✓ | 0.09 | 0.06 | 0.08 | 0.02 | 0.23 | 0.04 | 0.09 |
| Text Examples Patch | ✓ | 0.32 | 0.23 | 0.36 | 0.09 | 0.51 | 0.36 | **0.31** |
| **Idefics2** | | | | | | | | |
| No Context | - | 0.03 | 0.00 | 0.03 | 0.00 | 0.01 | 0.01 | 0.01 |
| Image Examples Prompt | × | 0.71 | 0.57 | 0.43 | 0.12 | 0.41 | 0.35 | 0.43 |
| Image Examples Patch | × | 0.58 | 0.32 | 0.40 | 0.03 | 0.39 | 0.17 | 0.31 |
| Text Examples Prompt | ✓ | 0.11 | 0.03 | 0.41 | 0.13 | 0.21 | 0.18 | 0.18 |
| Text Examples Patch | ✓ | 0.61 | 0.40 | 0.48 | 0.62 | 0.53 | 0.39 | **0.51** |

**Method.** In Figure 3a, we illustrate cross-modal patching. Given a task representation derived from one modality and query in another, the goal is to induce the model to output the correct task-specific answer. See Sec. 3.1 for all combinations of specification and query that we consider. For a task $t \in \mathcal{T}$ and model $f$, we run two forward passes: one to extract the task vector from the specification, and another to apply the vector onto an unseen query (see Figure 3a):

$$h_{l,\text{txt}}^t = f(p_{\text{txt}}^t) \qquad y_{\text{img}} = f_{\text{patch}}(x_{\text{img}} \mid h_{l,\text{txt}}^t) \quad (1)$$

Conditioned on the examples $p_{\text{txt}}^t$, the task vector $h_{l,\text{txt}}^t$ is extracted from the raw output of the $l$-th model layer at the delimiter token between the last query-answer pair, following standard practice in language-only studies (Hendel et al., 2023). For unseen query $x_{\text{img}}$, the task vector is then patched at the corresponding layer and token position of the query to induce the task-specific answer $y_{\text{img}}$, without otherwise specifying the task. Since the query modality is completely unseen, the representation needs to encode a very simple and generic function for the task to induce the correct output. In contrast with prior language-only studies (Hendel et al., 2023; Todd et al., 2024), our objective is to measure cross-modal alignment between task representations. Whereas these studies solely focus on text examples, we study additional modalities (images) and formats (instructions).

We compare cross-modal patching against *few-shot prompting* (Brown et al., 2020), where the task specification and query are jointly fed to the model (see Figure 3b). Few-shot prompting serves as a natural point of reference, as it is intervention-free and utilizes the full task information. However, these same characteristics also make it less informative than patching for analyzing the single task representation. In our experiments, we also refer to cross-modal patching and few-shot prompting as Patch and Prompt respectively.

## 4. Experimental Results

We start by evaluating cross-modal transfer. We quantify text-to-image transfer in Sec. 4.1, including LLM to VLM transfer in Sec. 4.2. We then examine instruction-based task vectors in Sec. 4.3. Finally, we analyze an extended set of VQA tasks in Sec. 4.4 and demonstrate how patching can override pre-existing tasks in Sec. 4.5.

**Models.** We consider three VLMs spanning both early and late-fusion architectures. LLaVA-v1.5 (Liu et al., 2024a) is a late-fusion model that fine-tunes a projection from visual features into the representation space of a language model. Mantis-Fuyu (Bavishi et al., 2023; Jiang et al., 2024) is an instruction-tuned variant of an early-fusion transformer trained to jointly handle image and text inputs from scratch, where the "visual encoder" is a linear projection on top of the raw image patches. Idefics2 (Laurençon et al., 2024) is a late-fusion model optimized for multimodal ICL.

**Implementation Details.** When conditioning on examples, we use the generic template from Todd et al. (2024):

$$\texttt{Q:}\{x_1\}\texttt{\textbackslash nA:}\{y_1\}\texttt{\textbackslash n\textbackslash n} \cdots \texttt{Q:}\{x_n\}\texttt{\textbackslash nA:}\{y_n\}$$

where we evaluate with $N = 5$ examples. For instructions, we pass the raw string with no templating. We evaluate on the six cross-modal tasks illustrated in Table 1, each split into a validation and test set. We determine the best layer to patch for each model via average task accuracy on the validation set. We report metrics on the unseen test set, averaged over three seeds. When computing accuracy metrics, we follow prior work (Hendel et al., 2023; Todd et al., 2024) and compare whether the first generated token is an exact match with the pre-defined label. We resize images to a standard width of 224 pixels. All qualitative examples correspond to Idefics2, the best performing model.

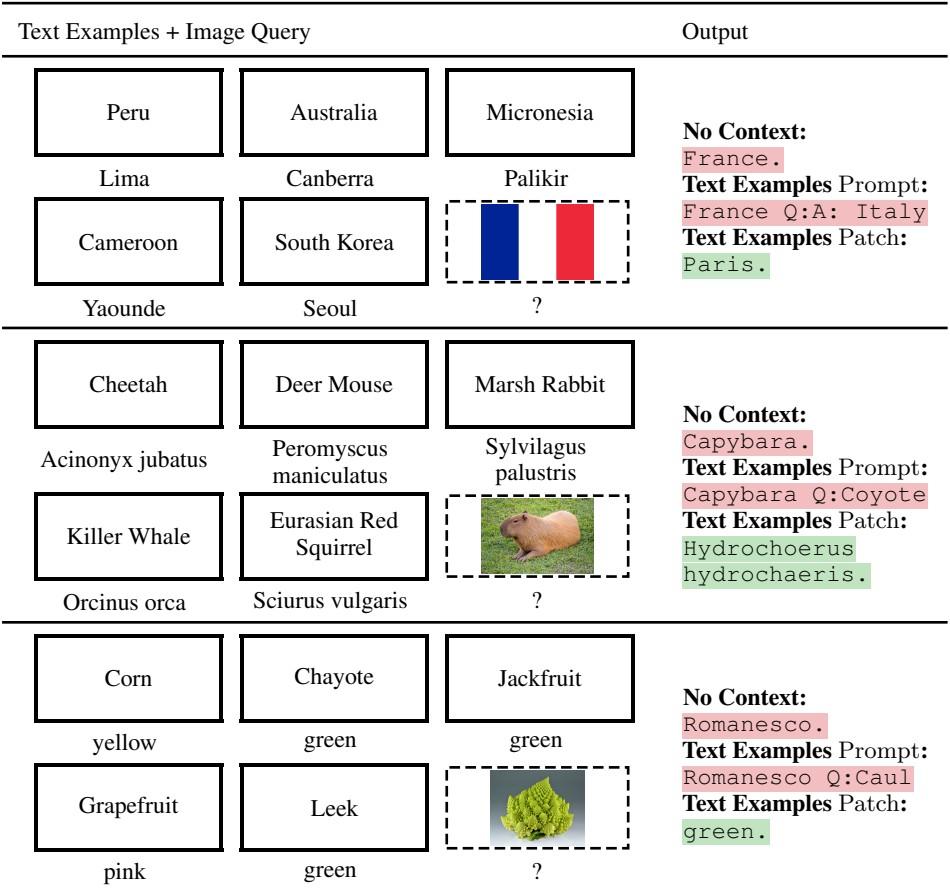

Figure 4: **Given the same text examples, patching is more effective than prompting.** We show qualitative examples transferring task information from text examples to image queries. Few-shot prompting (Prompt) regurgitates the input while cross-modal patching (Patch) successfully performs the task.

## 4.1. Transferring from Text Examples to Image Queries

**Setup.** We measure the performance of text examples applied to image queries on our six cross-modal tasks, following the same procedure illustrated in Figure 3. We evaluate our entire collection of early and late-fusion models, LLaVA-v1.5, Mantis-Fuyu, and Idefics2. We ablate two key axes of cross-modal patching (Text Examples Patch): the application method (Patch vs. Prompt) and specification modality (Text vs. Image Examples). We also provide the performance of two lower bounds – the majority answer from the examples (Random) and the query without any task information (No Context).

**Few-shot prompting struggles with cross-modality.** As seen in Table 2, few-shot prompting (Text Examples Prompt) struggles to execute the task on the image query across all models, performing at most 4% better than Random. In Figure 4, we depict a common failure mode of few-shot prompting, where the model "captions" or "regurgitates" the input, rather than adhering to the demonstrated pattern. In fact, the results are quite similar to the No Con-

text baseline, which may indicate that the model is ignoring the text preceding the image query. Since autoregressive VLMs are causally masked, there exists a task vector generated by the text examples in the prefix, but the model evidently struggles to correctly apply it to the image query. Patching resolves this issue by explicitly forcing the model to apply the task vector. Comparing Text Examples Patch to Text Examples Prompt, performance improves by 14-33% across all models (see Table 2).

**Text examples can outperform image examples.** The cross-modal text examples (Text Examples Patch) are more helpful than the unimodal image examples (Image Examples Prompt, Patch), outperforming the strongest unimodal baseline by 8-13% across all models. We hypothesize that image examples require an additional visual recognition step to understand the task compared with text examples, which may lead to noisier task representations (see Table 6).

> **Finding 1.** VLMs struggle with cross-modal few-shot prompting, which is fixed by cross-modal patching.

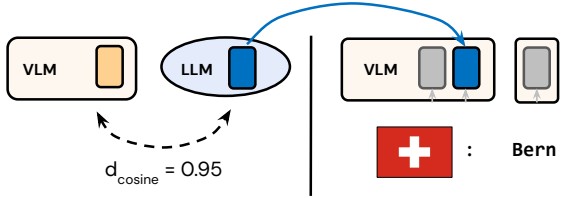

Figure 5: **Inter-model transfer.** For the same text examples, the base LLM and fine-tuned VLM contain highly similar task vectors (left). LLM task vectors can be patched onto image queries (right).

Table 3: **LLM to VLM transfer results**. We display the cosine similarity between the LLM and VLM task vectors as well as the test accuracy patching from text examples in the LLM to image queries in the VLM.

| Method | Avg. Cosine Sim. | Avg. Accuracy |
|---|---|---|
| Random | 0.58 | 0.14 |
| **LLaVA-v1.5** | | |
| VLM-VLM Patch | - | 0.32 |
| LLM-VLM Patch | 0.95 | **0.37** |
| **Idefics2** | | |
| VLM-VLM Patch | - | 0.51 |
| LLM-VLM Patch | 0.89 | **0.52** |

## 4.2. Transferring from LLMs to VLMs

**Setup.** Given that many VLMs are initialized from a pre-trained LLM, we explore the extent to which the task representations are preserved after fine-tuning. We limit this evaluation to late-fusion models with a corresponding LLM, where LLaVA-v1.5 corresponds to Vicuna (Chiang et al., 2023) and Idefics2 corresponds to Mistral (Jiang et al., 2023). For our six cross-modal tasks, we feed the same text examples to the LLM and VLM and compute the cosine similarity of the resulting task vectors. We also perform cross-modal patching from the LLM task vectors to image queries. We include a conceptual illustration in Figure 5.

**VLMs largely preserve LLM task representations.** The task vectors are highly similar across the base LLM and its fine-tuned VLM, with a cosine similarity of 0.89 or more (see Table 3). In contrast, the random baseline–the average similarity across all mismatched vector pairings–is 0.58.

**Pure language functions generalize to image queries.** In Table 3, we compare cross-modal patching in the inter-model case, with the VLM-only case copied from Table 2. Surprisingly, the inter-model setting performs 1-5% better than the VLM-only setting (LLM-VLM vs. VLM-VLM Patch). This result suggests VLMs can re-use functions learned only in language by LLMs, and that the base LLM's task representations are somewhat retained after fine-tuning.

> **Finding 2.** Task vectors can be patched from the base LLM to its corresponding VLM.

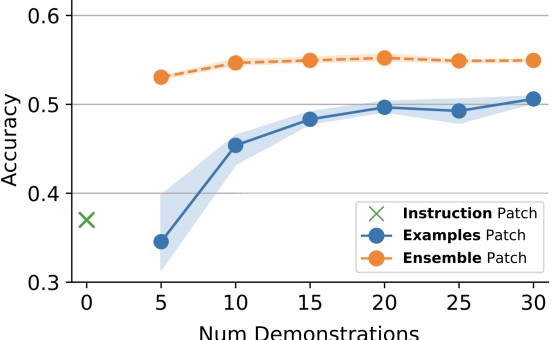

Figure 6: **Ensembling instruction- and example-based task vectors improves sample efficiency.** For cross-modal patching onto image queries, we compare the average task accuracy when using instructions, text examples, or an ensemble of the two. We plot the mean accuracy (solid lines) and variance (shaded regions), aggregated over three seeds.

## 4.3. Deriving Task Vectors From Instructions

**Setup.** Here, we demonstrate that task vectors can not only be defined with text examples but also instructions. We measure the cross-modal patching performance of Idefics2, averaged across our six tasks, for task vectors derived from instructions (Instruction Patch), examples (Examples Patch), or an ensemble of the two (Ensemble Patch). Ensembling refers to a simple element-wise average of the instruction- and example-based task vectors. We also analyze how performance scales with respect to the number of demonstrations. This highlights the contrast between instruction-based vectors, which require no data, and example-based vectors, which improve with more demonstrations. Because it is difficult to convey the desired casing style using instructions, in this section only we compute accuracy metrics in a case-insensitive fashion.

**Ensembling instructions and examples improves sample efficiency.** Here, we compare both approaches and assess their complementarity through ensembling. Instruction Patch shows competitive patching performance, matching that of Examples Patch composed of five samples (see Figure 6). Ensemble Patch performs even better, improving over the five-sample Examples Patch by 18%. Overall, combining the instruction-based vector improves the sample efficiency and reduces the variance of the example-based vector. We hypothesize that the ensemble performs well because the instruction provides a generic task definition less biased by the selection of examples while the examples clarify the output format.

> **Finding 3.** Beyond examples, task vectors can also be defined with instructions, which are more concise.

Table 4: **Cross-modal transfer on extended VQA tasks.** We show the test accuracy of cross-modal transfer on image queries for visual question answering tasks derived from VQAv2 (Goyal et al., 2017).

| Method | Food-Class | Shirt-Color | Man-Holding | Avg. |
|---|---|---|---|---|
| **Idefics2** | | | | |
| No Context | 0.00 | 0.00 | 0.00 | 0.00 |
| Image Examples Prompt | 0.70 | 0.41 | 0.46 | 0.52 |
| Image Examples Patch | 0.49 | 0.19 | 0.39 | 0.36 |
| Text Examples Prompt | 0.85 | 0.48 | 0.56 | 0.63 |
| Text Examples Patch | 0.93 | 0.56 | 0.59 | **0.69** |

Table 5: **Task overriding results.** Instruction Patch effectively steers the model to perform a newly introduced task.

| Method | Semantic | Syntax | Creative Generation | Factual Recall |
|---|---|---|---|---|
| Original Task | 0.10 | 0.15 | 0.08 | 0.15 |
| Original Task + System Prompt | 0.09 | 0.49 | 0.06 | 0.15 |
| Original Task + Instruction Patch | **0.36** | **0.59** | **0.65** | **0.42** |

## 4.4. Extended VQA Tasks

**Setup.** Beyond the synthetic tasks in our main evaluation set, we automatically construct an "in-the-wild" evaluation set derived from VQAv2 (Goyal et al., 2017), a visual question-answering dataset consisting of images and question-answer pairs. Recall that in our ICL setup, the model is only given input-output pairs and must infer the underlying task as a latent variable. However, VQAv2 encompasses a broad variety of tasks, from object recognition to color classification, which leads to ambiguities when treated as a single monolithic task. To overcome this issue, we use the questions to stratify samples into tasks. We curate a subset of questions asked across a large number of images, such that we can construct a 30-sample validation and 100-sample test set centered around the same task. To construct image examples, we drop the question and use the image-answer pairs as input-output pairs. To construct text examples, we use dense text descriptions generated by LLaVA-NeXT-34B from the LLaVA-ReCap dataset (Li et al., 2024) as each image's textual analog. We provide more details in Sec. A.1 of the Appendix.

**Patching can also be helpful for VQA tasks.** We report the results on these questions in Table 4, where we see that cross-modal patching (Text Examples Patch) results in a 6% improvement over few-shot prompting with text examples (Text Examples Prompt) and 17% improvement over few-shot prompting with image examples (Image Examples Prompt).

## 4.5. Overriding Pre-Existing Tasks

**Setup.** We now consider a special case of cross-modal patching where the task to patch conflicts with an existing task given in the prompt. This case mirrors a practical challenge where the user may request a task that goes against the global system instruction. In Figure 7, we show a few quali-

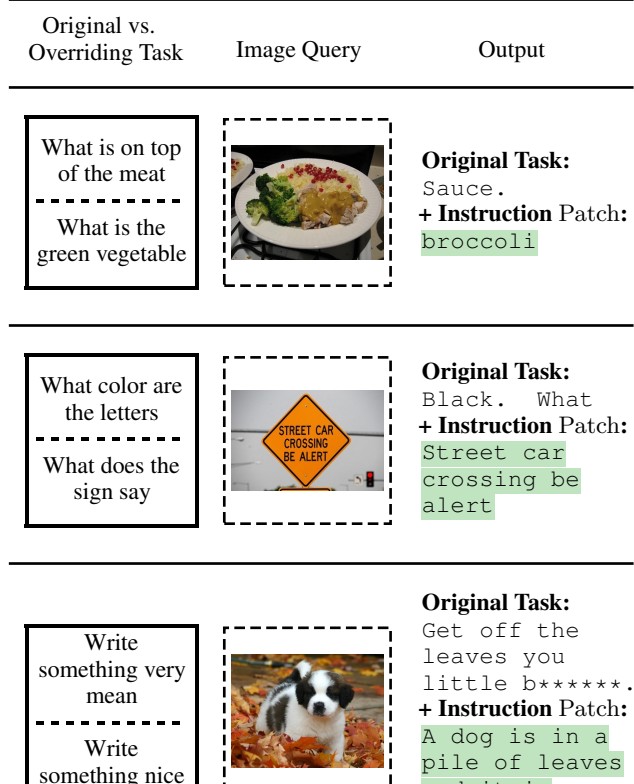

| Original vs. Overriding Task | Image Query | Output |
|---|---|---|
| What is on top of the meat / What is the green vegetable | | **Original Task:** `Sauce.` **+ Instruction** Patch: `broccoli` |
| What color are the letters / What does the sign say | | **Original Task:** `Black.  What` **+ Instruction** Patch: `Street car crossing be alert` |
| Write something very mean / Write something nice | | **Original Task:** `Get off the leaves you little b******.` **+ Instruction** Patch: `A dog is in a pile of leaves and it is adorable.` |

Figure 7: **Patching can override pre-existing tasks.** We show qualitative examples where the overriding task can supersede the original task when patched (Instruction Patch). Any offensive text has been redacted.

tative examples where this "overriding task," when patched as an instruction-based task vector, can supersede the "original task" in the prompt. We then construct an evaluation set stratified into four different settings for overriding.

1. *Semantic.* Both the original and overriding tasks query the image content. We randomly sample 1000 triplets of (image, original task, overriding task) from VQAv2 (Goyal et al., 2017).

2. *Syntax.* Both tasks refer to formatting instructions, i.e., answer in ALL CAPS, quotes, or JSON. We re-use the 1000 images from (1), now paired with randomly sampled conflicting syntax instructions.

3. *Creative Generation.* Both tasks are instructions for creative content, i.e., invent a book title, character name, or company name. We re-use the 1000 images from (1), now paired with randomly sampled conflicting creative instructions.

4. *Factual Recall.* Both tasks are image queries that require external knowledge. We use 148 overlapping images with conflicting questions from OK-VQA (Marino et al., 2019) and A-OKVQA (Schwenk et al., 2022).

Since some settings are open-ended, in this evaluation only, we use GPT4o (OpenAI, 2024) to automatically rate correctness, rather than exact string match against a pre-defined answer. We measure how often the generated answer indicates an intention to perform the overriding task. We compare cross-modal patching to the common alternative of including the instruction in the system prompt.

**Patching is more effective than system prompting.** As seen in Table 5, cross-modal patching is effective in steering the model toward a different task. For conflicts in Semantics, Creative Generation, and Factual Recall, patching outperforms system prompting by 27-59% (Instruction Patch vs. System Prompt). Interestingly, while system prompting performs extremely poorly in almost all settings, it performs much more competitively for the Syntax setting, likely because syntactic instructions are often not mutually exclusive. Nevertheless, for conflicts in Syntax, patching still outperforms system prompting by 10%.

## 5. Evolution of Shared Task Representations

Next, we analyze the answer generation process to understand how the VLM maps different modalities into shared representations. In Sec. 4.1 we use text examples as an alternative to image examples. To study why this is possible, we examine each modality independently, and compare the process in which answers are produced. We find that the representations evolve in a similar manner, despite the differences in modality. These representations arrive at an interpretable task vector that decodes into task summaries and clusters by task rather than modality.

**Setup.** For this analysis, we use Idefics2, which supports both text and image ICL. We refer to these settings as ICL because we do not apply any interventions, and therefore make no distinction between patching and prompting. We

condition the model on sets of text or image examples from our six tasks, then cache the representation of the last delimiter token across all model layers. To "decode" a representation, we normalize and project the activation with the model's unembedding matrix, which produces a probability distribution over all vocabulary tokens, following logit lens (nostalgebraist, 2020). Further details and visualizations can be found in Sec. A.8 of the Appendix.

**Text and image ICL induce three phases.** Here, we analyze the decodings across all model layers for the Country-Capital task. We see that early layers commonly decode to *auf* (which in Idefics2 globally corresponds to the colon), middle layers decode to task summaries like *headquarters*, and late layers decode to city names like *Rome* (see Figure 13 of the Appendix). This progression suggests that the last representation only encodes the input colon early on, compresses the ICL context into a task summary in the middle, then executes the task to produce the answer at the end. In Figure 8, we provide an overview of the layers where the phase changes from input to task to answer occur. For both text and image ICL, we observe that the input phase covers roughly the first 50% of layers, the task phase the next 40%, and the answer phase the last 10%. Hence, semantic representations emerge in the middle of the VLM, where task vectors derived from these layers achieve the best patching accuracy (see Figure 10 of the Appendix).

**The task phase decodes to task summaries.** In Table 6, we take a middle layer in the task phase (L=18) and depict the top decodings for all tasks. We find that task vectors defined in either modality often decode into meta-tokens that summarize the task, aligning with the observations made in text-only studies (Hendel et al., 2023; Todd et al., 2024). For example *headquarters*, *currency*, and *species* are the top-1 decodings for both text and image ICL in the first three tasks in the table. Even more, the decodings for image ICL are often noisier than text ICL, which suggests that cross-modal patching could help convey a cleaner expression of the task.

**The task phase clusters by task, not modality.** In Figure 2 we take a middle layer in the task phase (L=23) and co-embed the task vectors using t-SNE (van der Maaten & Hinton, 2008). Each point corresponds to a task vector derived from a single set of examples, which differ by modality (denoted by shape) and task (denoted by color). The ideal cross-modal representation space would group colors and intermix shapes; the task vectors largely cluster according to this structure (see Figure 2b). While all other tasks form distinct groups, we hypothesize that the food-related tasks (Food-Color; purple and Food-Flavor; brown) are not well-separated because food color and flavor are correlated. In contrast, for the same layer, the image and text embeddings in the context summarized by the task vector are much more modality-sensitive (Figure 2a). The sensitivity of the context

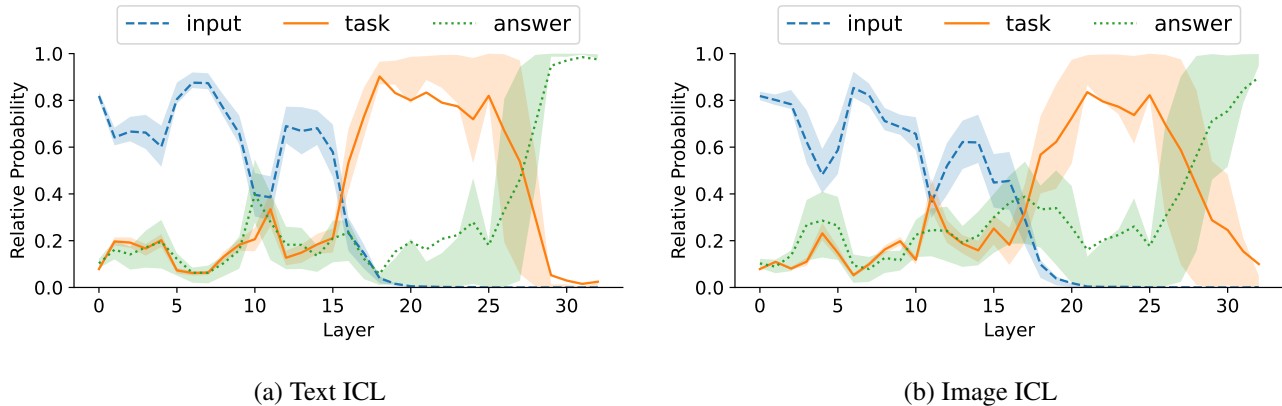

(a) Text ICL                                        (b) Image ICL

Figure 8: **The output evolves in three distinct phases that are shared for text and image ICL**. Each line represents the relative probability that the layer representation decodes to one of the three pre-defined tokens corresponding to the input, task, and answer. We plot the mean probability (solid lines) and variance (shaded regions), aggregated over 100 sets of examples. We visualize only the Country-Capital task; all tasks are in Figure 14 of the Appendix.

Table 6: **Task vectors, whether textual or visual, often decode to task summaries.** The table depicts the top-5 decodings for each task, where ◇ denotes non-word tokens.

| Task | Text ICL | Image ICL |
|---|---|---|
| Country-Capital | *headquarters, cities, city, cidade, centro* | *headquarters, administr, cities, city,* ◇ |
| Country-Currency | *currency, currency, dollar, dollars, Currency* | *currency,* ◇*, currency, undefined, dollars* |
| Animal-Latin | *species, genus, habitat, mamm, american* | *species, genus, mamm, spec, creature* |
| Animal-Young | *pup, babies, baby, called, young* | *young, species, script-style, animal, teenager* |
| Food-Color | *yellow, pink, green, purple, orange* | *green, yes, yellow, verd, yes* |
| Food-Flavor | *flavor, taste, mild, flav, tastes* | *yes, none, anger, cerca, vegetables* |

embeddings is consistent with Liang et al. (2024), which observes distinct clustering by modality throughout model layers in VLMs. This dichotomy suggests that the VLM maintains modality-specific embeddings in the context and integrates modalities in the task vector.

> **Finding 4.** VLMs map text and image examples, as well as instructions, to similar task vectors, despite differences between the image or text embeddings in the context.

## 6. Limitations

In this work, we show that VLMs learn cross-modal task representations but lack a definitive explanation for *why*. Empirical studies offer several hypotheses, such as the existence of isomorphic structures between language and other perceptual representation spaces (Abdou et al., 2021; Patel

& Pavlick, 2022; Pavlick, 2023), or model and data scale as drivers for representational convergence (Huh et al., 2024).

## 7. Conclusion

Despite the success of autoregressive VLMs, we lack a clear understanding of their hidden task representations. Our primary observation is that VLMs map inputs into a shared task representation space, regardless of whether the task is defined by text examples, image examples, or explicit instructions. We evaluate cross-modal transfer, where we observe that few-shot prompting can struggle with input regurgitation while patching induces the task more effectively. We show that task vectors can be transferred from a base LLM to a fine-tuned VLM, possibly indicating representational re-use of functions learned in language. We also show that task vectors can be defined with instructions, which improves the sample efficiency of example-based task vectors. We hope that our work will inspire further analysis of shared representation spaces and the internals of VLMs.

## Acknowledgements

We would like to thank Jiahai Feng, Stephanie Fu, Alexander Pan, Alberto Hojel, Lisa Dunlap, Chung Min Kim, Brent Yi, Candace Ross, and Koustuv Sinha for helpful discussions and feedback on the paper. Authors were supported in part by the NSF, DoD, and/or the Berkeley Artificial Intelligence Research (BAIR) industrial alliance program.

## Impact Statement

This paper studies the cross-modal properties of task vectors. This result yields promising implications for accessibility and interactivity – users are no longer limited to unimodal examples and can be more expressive when communicating tasks to VLMs. However, the reliability of these models still remain a limitation. While VLMs can readily adapt to newly introduced tasks, this often comes at the cost of task accuracy. Further stress testing is advised before deployment in critical applications.

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

# A. Appendix

## A.1. Experimental Details

**Models.** We provide further details on the models used in our evaluation in Table 8.

**Tasks.** We show representative examples in Table 1. We scrape the images for all tasks from Wikipedia, because we find that the images tend to depict more clearly identifiable prototypes, unlike traditional computer vision datasets. For some tasks the labels were automatically generated by Claude 3.5 Sonnet (Anthropic, 2024) and manually cross-checked, unless otherwise noted.

- **Country-Capital**. Given the name of the country or its flag, predict the capital city. The text-only case is identical to Todd et al. (2024).

- **Country-Currency**. Given the name of the country or its flag, predict the official currency. The text-only case is almost identical to Todd et al. (2024), except we remove the country modifier from the currency to make the task harder.

- **Animal-Latin**. Given the name of the animal or its image, predict its scientific name in Latin. The labels are derived from the mammals categorized in iNaturalist (2021).

- **Animal-Young**. Given the name of the animal or its image, predict the term for its baby.

- **Food-Color**. Given the name of a fruit or vegetable or its image, predict its iconic color. This task is inspired by the conceptual example first proposed in Hendel et al. (2023).

- **Food-Flavor**. Given the name of a fruit or vegetable or its image, predict its iconic flavor profile.

**Extended VQA Tasks.** Here we provide additional details on the tasks evaluated in Sec. 4.4.

- **Food-Class**. Given the image or description, answers: *What kind of food is this?*

- **Shirt-Color**. Given the image or description, answers: *What color is the man's shirt?*

- **Man-Holding**. Given the image or description, answers: *What is the man holding?*

## A.2. Computational Overhead

We report the computational overhead of cross-modal patching in Table 7. Unlike few-shot prompting which must retain the examples in the context, patching replaces them with a single activation, which significantly reduces the computational cost. Since this effect is most apparent for long contexts, we evaluate on text examples with dense descriptions (see Sec. 4.4). Patching reduces runtime by 11x and

VRAM consumption by 2.4x when compared with prompting. The cost of patching is almost equivalent to processing the query only, since the VLM no longer needs to attend to the long context. Note that computing the task vector requires an upfront cost similar to few-shot prompting, but it is amortized in future runs – unlike prompting which requires re-processing the context each time.

Table 7: We report the computational overhead of a forward pass on N=30 Text Examples, averaged over 100 runs.

| Method | Runtime (seconds) | VRAM (GB) |
|---|---|---|
| Prompting (Context + Query) | 2.20 | 20.02 |
| Patching (Task Vector + Query) | 0.20 | 8.21 |
| Query Only | 0.19 | 8.21 |

## A.3. Robustness to Noisy Instructions

In Table 9 we examine cross-modal patching's robustness to noisy instructions. We evaluate on the same instructions and images described in Table 1, except we introduce typos to the instructions by randomly swapping consecutive characters within words, following the protocol of Kumar (2023). As expected, the performance degrades as the number of typos increases. However, even with typos, patching is able to maintain non-negligible performance.

## A.4. Task Overriding Details

We provide further details for the experimental setup of Sec. 4.5. We use GPT4o to automatically rate correctness, which we further illustrate in Figure 9. Below, we also provide example tasks for each setting. Note that for Syntax and Creative Generation, we show the full pool of tasks below. For Semantic and Factual Recall, the examples are representative of a larger pool of questions.

1. Semantic
   - *What color is the umbrella?*
   - *What does the sign say?*
   - *What is in the bowl?*
2. Syntax
   - *Format your answer in ALL CAPS.*
   - *Put your answer in quotes.*
   - *Answer in JSON format.*
3. Creative Generation
   - *Write a creative book title that matches the image.*
   - *Invent a name for the main character of the image.*
   - *Name a company that could use this image in an advertisement.*
4. Factual Recall
   - *Which country won the 2018 world cup?*
   - *From which culture does this food originate?*
   - *How was this valley formed?*

Table 8: We study a diverse set of representative VLMs spanning both early-fusion and late-fusion paradigms and varying image ICL capabilities.

|  | LLaVA-v1.5 (Liu et al., 2023a) | Mantis-Fuyu (Jiang et al., 2024) | Idefics2 (Laurençon et al., 2024) |
|---|---|---|---|
| Text Model | Vicuna (Chiang et al., 2023) | Fuyu (Bavishi et al., 2023) | Mistral (Jiang et al., 2023) |
| Vision Model | CLIP (Radford et al., 2019) | Fuyu (Bavishi et al., 2023) | SigLIP (Zhai et al., 2023) |
| Paradigm | Late-Fusion | Early-Fusion | Late-Fusion |
| Image ICL | No | Yes | Yes |
| Parameters | 7B | 8B | 8B |
| Layers | 32 | 36 | 32 |

Given the model answer, determine whether it was trying to answer:
A. 'question1'
B. 'question2'
C. Both
D. Neither

It does not matter if the answer is ungrammatical or cut off; assume the model's intent to sort its answer into the right category. Output your rating as a JSON containing the key 'choice' corresponding to each category.

```
answer: <answer>
question1: <question1>
question2: <question2>
output:
```

Figure 9: We prompt to GPT4o to automatically rate task overriding. We include case "C" because the VLM sometimes generates multiple sentences to cover both tasks, which we count as a success case along with case "B."

Table 9: We report the accuracy of cross-modal patching of noisy instructions onto image queries.

| Num. Character Swaps | Country-Capital | Country-Currency | Animal-Latin | Animal-Young | Food-Color | Food-Flavor | Avg |
|---|---|---|---|---|---|---|---|
| s=0 | 0.58 | 0.22 | 0.34 | 0.44 | 0.48 | 0.29 | 0.39 |
| s=1 | 0.65 | 0.07 | 0.33 | 0.51 | 0.52 | 0.13 | 0.37 |
| s=2 | 0.63 | 0.14 | 0.36 | 0.41 | 0.48 | 0.08 | 0.35 |

## A.5. Extended Discussion of Text Example Transfer

**Template Format.** While in our main experiments we use the generic template proposed by Todd et al. (2024), here we use the model-specific template for Idefics2:

$$\text{User:}\{x_1\}\texttt{<end\_of\_utterance>}\texttt{\textbackslash nAssistant:}\{y_1\}$$

As seen in Table 10, the trends in performance remain consistent with Table 2 – cross-modal patching significantly outperforms few-shot prompting with text examples.

**LLM to VLM Transfer.** Corresponding to Table 3, in Table 11 we display extended LLM-VLM transfer results.

**Validation Performance.** In our main experiments, we present the test performance of a single model layer, which achieves the best average task accuracy on the validation set. In Figure 10 we show the performance of all model layers on this validation set. For the late-fusion models, the best task vector lies near the middle of the network. In contrast,

for the early-fusion model, the best task vector lies in the late-middle layers. When comparing tasks, the shape of the curve tends to fall into two categories: a peak then plateau (Food-Color, Food-Flavor) or single sharp peak (all other tasks). We hypothesize that the shape is associated with the diversity of the output space – fewer possible outputs make it more likely for later layers, which are closer to the answer representation, to yield a plausible result.

## A.6. Ablating All Modality Combinations

In Table 12, we display additional results when patching task vectors in all combinations of example-query modality. For image queries, the cross-modal setting is highly beneficial, where task vectors derived from text examples outperform those from image examples by 11-20% respectively. For text queries, this is not the case, where the cross-modal setting underperforms by 9-23%.

Table 10: We ablate the template format and display the test accuracy when transferring from text ICL to image queries. We use the recommended template for Idefics2.

| Method | Country-Capital | Country-Currency | Animal-Latin | Animal-Young | Food-Color | Food-Flavor | Avg. |
|---|---|---|---|---|---|---|---|
| **Idefics2** | | | | | | | |
| No Context | 0.00 | 0.00 | 0.07 | 0.00 | 0.00 | 0.00 | 0.01 |
| Image Examples Prompt | 0.74 | 0.53 | 0.44 | 0.12 | 0.43 | 0.35 | 0.44 |
| Image Examples Patch | 0.78 | 0.09 | 0.40 | 0.02 | 0.02 | 0.01 | 0.22 |
| Text Examples Prompt | 0.16 | 0.06 | 0.24 | 0.16 | 0.17 | 0.12 | 0.15 |
| Text Examples Patch | 0.70 | 0.44 | 0.50 | 0.64 | 0.54 | 0.40 | **0.54** |

Table 11: We show the test accuracy when transferring task vectors from text ICL in the LLM to image queries in the VLM.

| Method | Country-Capital | Country-Currency | Animal-Latin | Animal-Young | Food-Color | Food-Flavor | Avg. |
|---|---|---|---|---|---|---|---|
| **LLaVA-v1.5** | | | | | | | |
| VLM-VLM Patch | 0.31 | 0.30 | 0.26 | 0.18 | 0.53 | 0.31 | 0.32 |
| LLM-VLM Patch | 0.33 | 0.32 | 0.25 | 0.33 | 0.53 | 0.45 | **0.37** |
| **Idefics2** | | | | | | | |
| VLM-VLM Patch | 0.61 | 0.40 | 0.48 | 0.62 | 0.53 | 0.39 | 0.51 |
| LLM-VLM Patch | 0.57 | 0.58 | 0.46 | 0.55 | 0.54 | 0.39 | **0.52** |

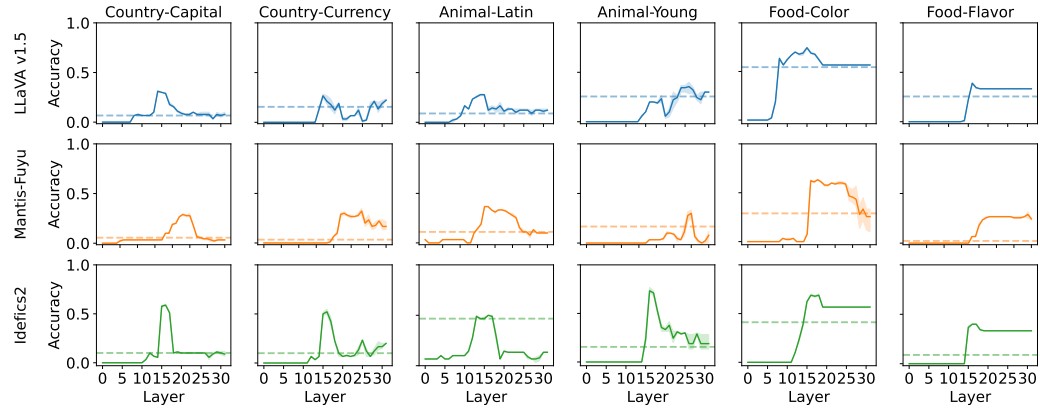

Figure 10: We display validation performance for transferring task vectors from text examples to image queries across model-task combinations. Each subplot shows the accuracy by model layer, with a dotted line providing the few-shot prompting baseline accuracy for reference.

Table 12: We display the test accuracy when patching task vectors in all combinations of example-query modality. The best-performing combination for a given query modality is highlighted. Each setting is denoted as {Specification Modality}-{Query Modality}. The best-performing combination for a given query modality is highlighted.

| Method | Country-Capital | Country-Currency | Animal-Latin | Animal-Young | Food-Color | Food-Flavor | Avg. |
|---|---|---|---|---|---|---|---|
| **LLaVA-v1.5** | | | | | | | |
| Image - Image Patch | - | - | - | - | - | - | - |
| Text - Image Patch | 0.31 | 0.30 | 0.26 | 0.18 | 0.53 | 0.31 | 0.32 |
| Text - Text Patch | 0.97 | 0.58 | 0.77 | 0.20 | 0.63 | 0.41 | 0.59 |
| Image - Text Patch | - | - | - | - | - | - | - |
| **Mantis-Fuyu** | | | | | | | |
| Image - Image Patch | 0.17 | 0.03 | 0.16 | 0.05 | 0.50 | 0.31 | 0.20 |
| Text - Image Patch | 0.32 | 0.23 | 0.36 | 0.09 | 0.51 | 0.36 | 0.31 |
| Text - Text Patch | 0.46 | 0.30 | 0.48 | 0.18 | 0.28 | 0.36 | 0.34 |
| Image - Text Patch | 0.31 | 0.01 | 0.36 | 0.05 | 0.40 | 0.34 | 0.25 |
| **Idefics2** | | | | | | | |
| Image - Image Patch | 0.58 | 0.32 | 0.40 | 0.03 | 0.39 | 0.17 | 0.31 |
| Text - Image Patch | 0.61 | 0.40 | 0.48 | 0.62 | 0.53 | 0.39 | 0.51 |
| Text - Text Patch | 0.97 | 0.61 | 0.74 | 0.54 | 0.63 | 0.41 | 0.65 |
| Image - Text Patch | 0.81 | 0.43 | 0.58 | 0.04 | 0.40 | 0.27 | 0.42 |

### A.7. Transferring from Image Examples to Text Queries

Here we assess the usefulness of task vectors derived from image examples for text queries. In Figure 11 we depict a set of tasks that involve recognizing visual concepts in dense textual descriptions, including mapping the description to a technology company, cartoon character, or popular meme.

Similar to Sec. 4.1, the model struggles when cross-modal examples are applied via few-shot prompting (Image Examples Prompt) but performs well when the same examples are patched as a task vector (Image Examples Patch). Both baselines (Text Examples Prompt, Image Examples Prompt) sometimes generate incorrect answers within the same output domain, suggesting that, rather than focusing on the input-output relationship, the model may be ignoring the input image or description. However, on the evaluation tasks in Table 2, it is difficult for image examples to surpass the unimodal text baselines. In Table 12 of the Appendix we include an ablation containing all possible combinations of specification-query modality for task vector patching, where text examples consistently outperform image examples regardless of the query modality. We hypothesize that this phenomenon can be attributed to the nature of the tasks themselves. In the evaluation tasks, when conditioned on image examples the model also has to complete an implicit recognition task mapping the image to the underlying textual concept. For example, if the model cannot match the flag to the correct country name, it will not be able to predict the correct currency. However, if recognition is instead required in text space, as is the case in Figure 11, image examples may better encode the task. We think that the curation of a comprehensive evaluation set containing dense text descriptions and corresponding visual concepts is an exciting future direction.

**Dense Text Descriptions.** Corresponding to Figure 11, we display the text descriptions used in the text examples.

- {*The logo is a rainbow-colored apple.* : **Apple**}
- {*The logo is a white ghost against a yellow background.* : **Snapchat**}
- {*The logo is a white camera against a gradient background.* : **Instagram**}
- {*The logo is the letter P stylized to look like a pushpin.* : **Pinterest**}
- {*The character is a squirrel wearing an astronaut suit.* : **Sandy Cheeks**}
- {*The character is a puffer fish wearing a blue shirt, red skirt, and blue hat.* : **Mrs. Puff**}
- {*The character is a crab wearing a blue shirt, blue pants, and brown belt.* : **Mr. Krabs**}
- {*The character is a pink starfish wearing green and purple pants.* : **Patrick Star**}
- {*An image of an orange and white cat wearing a blue shirt playing the keyboard.* : **Keyboard Cat**}

- {*An image of a shiba inu sitting on a couch.* : **Doge**}
- {*A cartoon of a dog wearing a hat sitting in a room engulfed with flames.* : **This Is Fine Dog**}
- {*An image of an unhappy cat with blue eyes and white and brown fur.* : **Grumpy Cat**}

### A.8. Representation Evolution For All Tasks

**Implementation Details.** For this experiment, we condition the model on some task specification (e.g., text ICL, image ICL) and cache the intermediate activation across all model layers. We "decode" the representations using logit lens (nostalgebraist, 2020), which produces a probability distribution over all vocabulary tokens. For each task and specification type, we aggregate statistics over 100 runs with different sets of $N = 5$ examples.

**Discrete Visualization.** In our discrete visualization (Figure 13), we collect the top-1 token from each run and visualize it as a slice in a pie chart. Below each pie chart, we also show the token corresponding to the largest slices, which represents the most common decodings across all runs.

**Continuous Visualization.** We provide the pseudocode for the continuous visualization (Figure 14) in Figure 12. In this visualization, we compare the relative probability of three pre-defined tokens corresponding to the input, task, and answer. Specifically, we use the token *auf* for the input, one of {*capital, currency, species, baby, color, flavor*} for the task, and each run's ground-truth label for the answer. For each run, we take the softmax of the three token probabilities to obtain a normalized probability distribution.

**t-SNE Visualization.** For the visualizations in Figure 2, we co-embed representations from the same middle layer using t-SNE (van der Maaten & Hinton, 2008). Specifically, we take the raw layer activations, normalize them using the model's final normalization layer to ensure they are comparable in scale, and then apply the t-SNE algorithm. We select a random subset of 30 points for each (task, modality) combination. For the "context" embeddings, we take the embeddings corresponding to the input in the input-output pairs, excluding any tokens related to the template or output. For image ICL, this refers to the embeddings of the input image, while for text ICL, it refers to the embeddings of the input text. For the "task vector" embeddings, we take the embedding corresponding to the last delimiter token, which summarizes the preceding context.

**Conditioning on Instructions.** We visualize the token representation evolution when conditioning on instructions rather than examples in Figure 15 and Figure 13. We do not display discrete pie charts since a single instruction does not produce aggregate statistics, unlike examples where there are multiple possible sets. The instruction-based vector decodings are often interpretable and resemble a meta summary for the task, similar to the observations in Sec. 5.

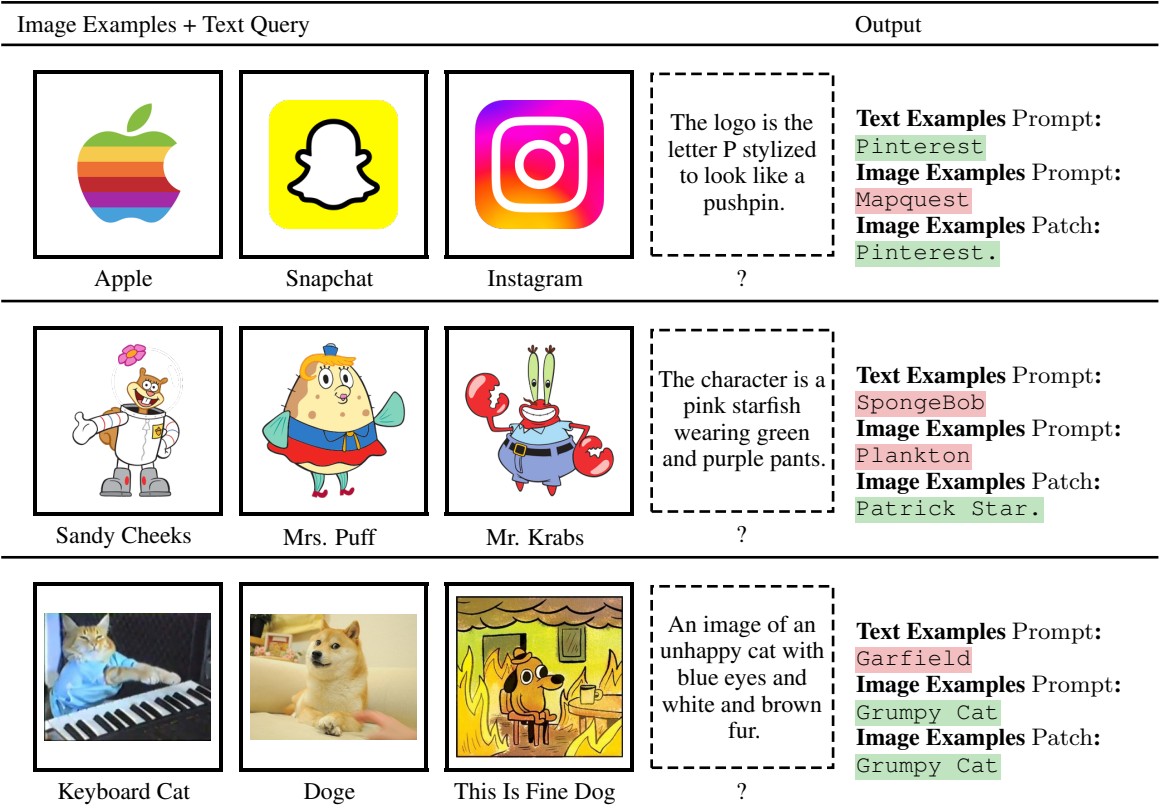

Figure 11: **Cross-modal transfer from image examples to text queries**. We show qualitative examples where few-shot prompting with text examples (Prompt) and image examples (Prompt) often produces incorrect predictions in the same output domain while cross-modal patching (Patch) leads to the correct answer.

```python
def continuous_rep_evolution(model, dataset, select_vocab_idx):
    """
    Plots the relative probability of the input, task, and answer token (given by
    `select_vocab_idx`) across layers of `model`, for a `dataset` representing a task.
    """
    dataset_rel_prob = []
    for sample in dataset:
        # dim is [num_layers, 1, hidden_dim]
        feats = cache_act(model(sample))
        feats = model.norm(feats)
        # dim is [num_layers, 1, vocab_size]
        token_dist = model.lm_head(feats)
        input_idx, task_idx, answer_idx = select_vocab_idx
        # dim is [num_layers, 1, 3]
        token_dist = token_dist[:, :, [input_idx, task_idx, answer_idx]]
        rel_prob = softmax(token_dist)
        dataset_rel_prob.append(rel_prob)
    plot_layer_vs_rel_prob(dataset_rel_prob)
```

Figure 12: PyTorch-like pseudocode for the continuous visualization shown in Figure 14.

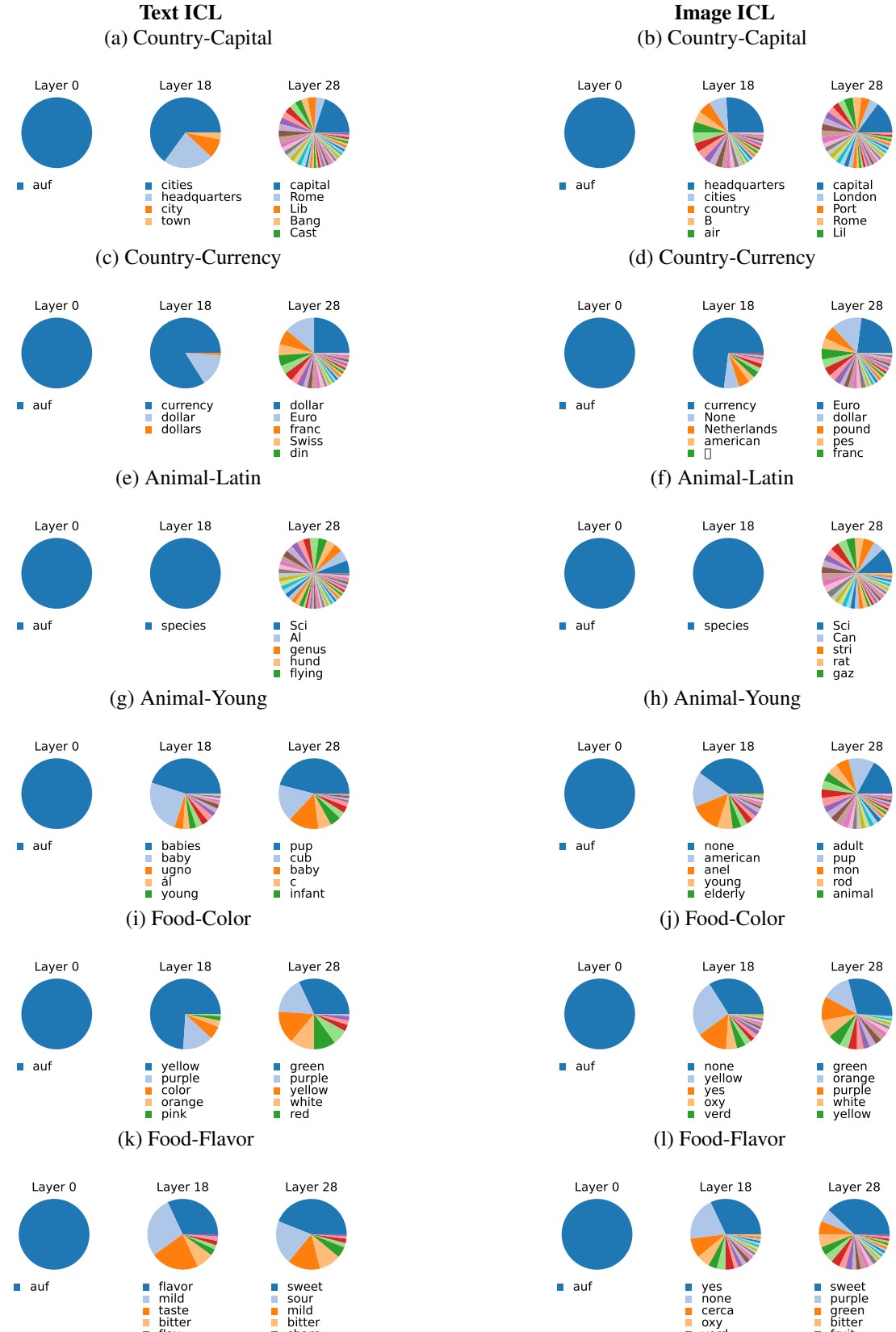

Figure 13: We show a discrete visualization of how the token representation evolves across layers for all tasks. Each pie chart slice represents a top-1 decoding across 100 sets of examples, and the most common decodings are displayed below.

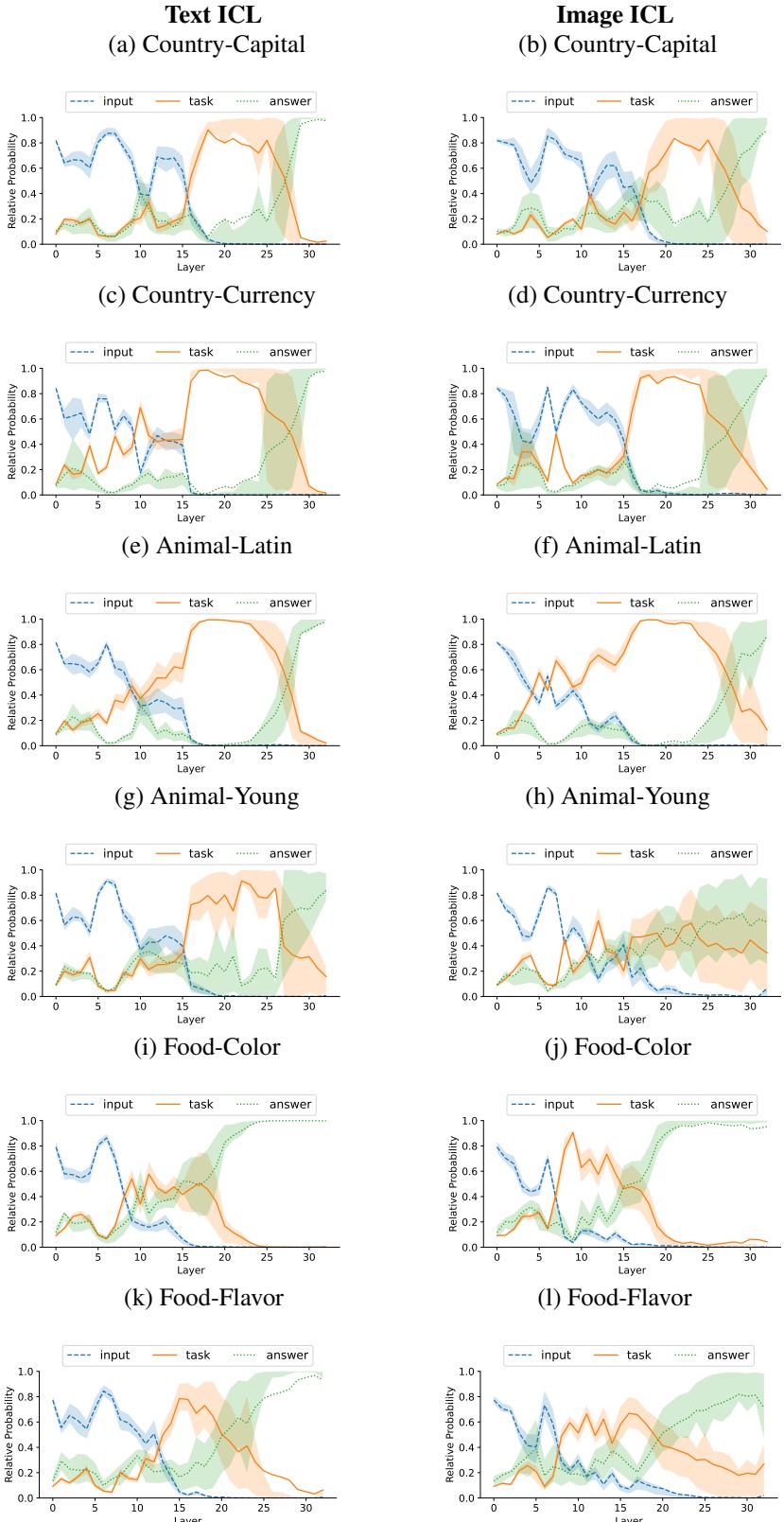

Figure 14: We show a continuous visualization of how the token representation evolves across layers for all tasks. We use the token *auf* for the input, one of {*capital, currency, species, baby, color, flavor*} for the task, and each run's ground-truth label for the answer. See Figure 12 for the relevant pseudocode.

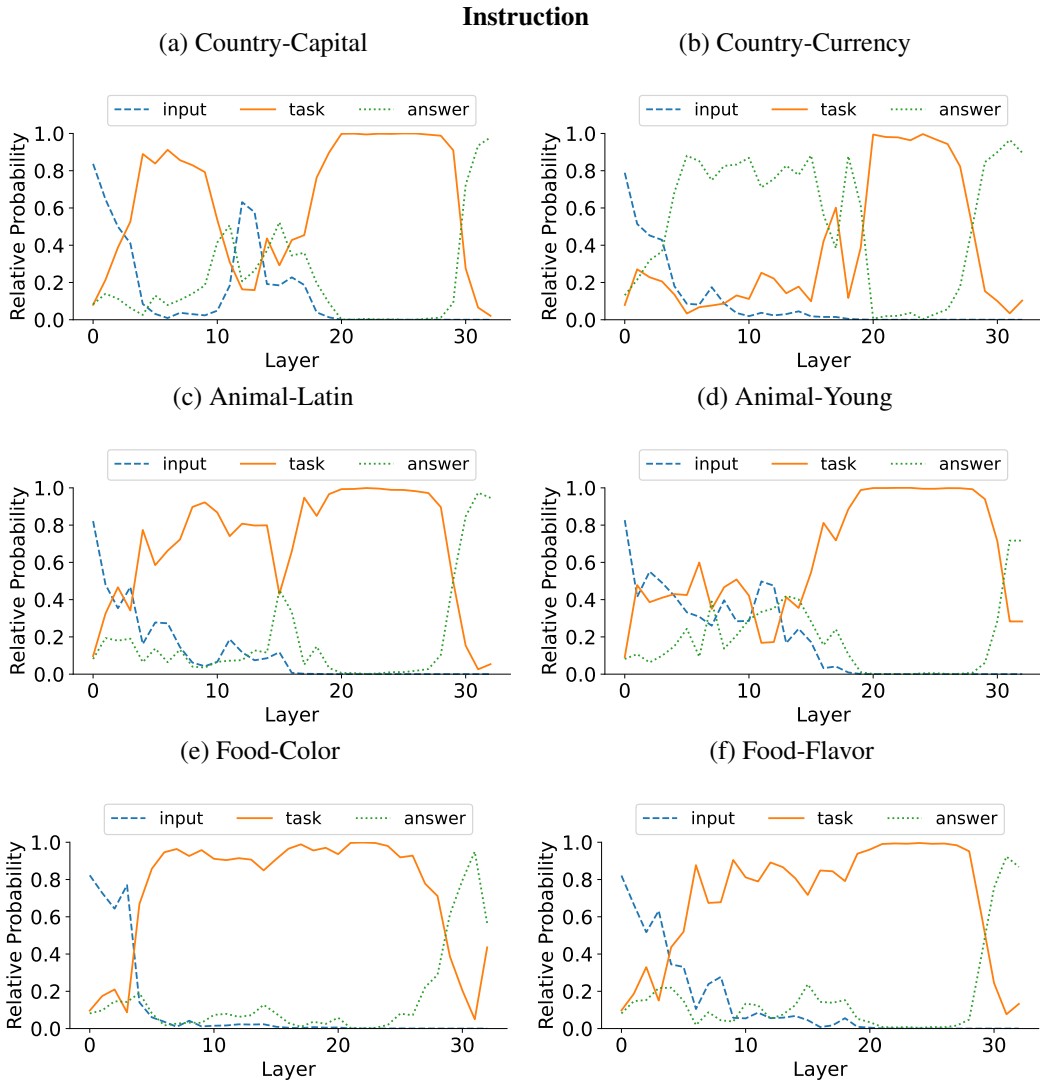

Figure 15: We show a continuous visualization of the token representation evolution when conditioned on instructions rather than examples. The results are aggregated over a single instruction rather than multiple examples, so there are no variance bars.

Table 13: We depict the top-5 decodings for the instruction-based vector, where ◊ denotes symbols that do not correspond to common word tokens.

| Task | Instruction |
|---|---|
| Country-Capital | *city, GU, vik, cities, headquarters* |
| Country-Currency | *◊, ◊, ◊, itos, ◊* |
| Animal-Latin | *species, genus, ◊, animals, american* |
| Animal-Young | *baby, babies, ◊, bach, called* |
| Food-Color | *colors, color, colour, ETH, ilo* |
| Food-Flavor | *taste, tastes, arom, food, flavor* |

