# OpenReview forum: "Vision-Language Models Create Cross-Modal Task Representations"
_ICML.cc/2025/Conference — ICML 2025 poster_

### Official Review · Reviewer_6RYy · 2025-03-08

**Overall Recommendation:** 3

**Summary:**

This paper examines a phenomenon in VLMs, where they encode inputs into a unified representation space, regardless of whether the task is defined through text examples, image examples, or explicit instructions. Building on this, the authors conduct experiments to assess the model's cross-modal transfer capability in an in-context learning setting.

## update after rebuttal
I have read the materials provided by the authors. The experiments on more popular VQA benchmarks are obviously less comprehensive. Therefore, I decided to keep my original rating (weak accept).

**Claims And Evidence:**

The paper presents sufficient evidence supporting its main argument: the presence of a shared task representation space in VLMs. This is evident, as visual and textual tokens are processed within the same representation space in LLMs. Additionally, experimental results demonstrate that this alignment persists across different network layers.

**Essential References Not Discussed:**

No.

**Experimental Designs Or Analyses:**

Yes. Table 2, Table 3, Figure 6, Table 4, Figure 10. They all look good to me, showcasing that the visual and textual tokens are aligned in LLM and VLM.

**Methods And Evaluation Criteria:**

I think the evaluation criteria are proper. Using in-context learning to identify the similarity in representation space.

**Other Comments Or Suggestions:**

If the authors can determine the implications of this observation for the VLM community—such as its potential to enhance the development of more effective or efficient VLM models and algorithms—their findings would carry significantly greater impact. Understanding how this shared representation space can be leveraged to improve model performance, reduce computational costs, or enhance cross-modal learning could lead to meaningful advancements in the field. Furthermore, exploring its applications in real-world tasks, such as multimodal reasoning, retrieval, or generation, would further solidify its practical significance.

**Other Strengths And Weaknesses:**

Strength:
- The insight of the paper is interesting and the paper is well-written.
- Extensive experiments have been done to verify the claim of the paper.

Weakness:
- The core argument is somewhat less surprising, as it is straightforward that VLM maps inputs from different modalities into a shared space.

**Questions For Authors:**

- How can we learn from this finding when building visual-language cross-modality models?

**Relation To Broader Scientific Literature:**

This paper can be referred to by papers building new VLM.

**Theoretical Claims:**

No theory was proposed in the paper.

---

> ### Author Rebuttal · Authors · 2025-04-01
>
> Thank you for your valuable comments. Below, we include results regarding our core argument and our method’s computational cost, following your suggestions.
>
> ---
>
> > 1. The core argument is somewhat less surprising, as it is straightforward that VLM maps inputs from different modalities into a shared space.
>
> We study VLMs trained to map image embeddings into the representation space of an LLM.
> - Although this can be simplified as mapping different modalities into a shared space, this perspective does not explain our results.
> - **In fact, Figure 2a of the main text shows that the image and text embeddings don’t cluster by task. In contrast, Figure 2b shows that the task vector at the end of the sequence does group by task.**
> - This grouping in Figure 2b is not supervised in any way – its emergence is driven only by the next token prediction loss, or the fact that the answer is conceptually similar.
> - We find it striking that these task representations, learned without explicit supervision, are aligned across modalities (image, text) and specifications (examples, instructions). We believe these are new and valuable insights that can inform future research.
>
> ---
>
> > 2. How can we learn from this finding when building visual-language cross-modality models?
>
> **First, our method reduces computational cost, especially for long contexts.** In Table 4, we show that in practice, patching reduces runtime by 11x and VRAM consumption by 2.4x when compared with few-shot prompting, for long text descriptions (on the dataset from Sec. A.4 of the main text). This is because the VLM no longer needs to attend to the long context, after injecting the single task vector, making the cost effectively equivalent to processing the query only.
>
> **Second, our LLM to VLM transfer experiments reveal gaps in model training.** In Table 3 of the main text, we see that for the same text inputs, the VLM produces lower-quality task vectors than the LLM, as evidenced by the 1-5\% performance degradation across multiple models. One could apply this insight to VLM training by introducing a cosine similarity loss between the LLM and VLM task vectors on language-only examples, to monitor the observed degradation in language capabilities.
>
> We will be sure to add further discussion of these implications of our findings to the final manuscript.
>
> *Table 1. Computational overhead of patching. Overhead of a single forward pass on N=30 Text ICL examples, averaged over 100 runs.*
> | Method          | Runtime (seconds) | VRAM (GB) |
> |----------------------|-------------------|-----------|
> | Prompting (Context + Query)      | 2.20             | 20.02   |
> | Patching (Task Vector + Query)  | 0.20             | 8.21     |
> | Query Only           | 0.19             | 8.21     |

---

> > ### Comment · Reviewer_6RYy · 2025-04-05
> >
> > I would like to thank the authors for further explanation. Patching is used only for in-context learning, but not for more practical VQA tasks in the experiments. Therefore, my concerns remain, and I decide to maintain my original ratings.

---

> > > ### Author Response · Authors · 2025-04-05
> > >
> > > Thank you for your comment, and thank you for taking the time to give feedback on our paper!
> > >
> > > **In Sec. A.4 of the Appendix, we include an evaluation based on VQAv2.** We copy the results to Table 2 below, showing that patching is also effective in more practical VQA tasks. Following this discussion, we will move this result to the main text. [We also invite you to view Table 2 in our response to Reviewer CQRE](https://openreview.net/forum?id=77ziPGdQct&noteId=fxPChMsZmQ), which shows that patching performs well at task overriding on four new VQA settings, including questions from VQAv2 [1], OKVQA [2], and A-OKVQA [3].
> > >
> > > *Table 2. We show the test accuracy of cross-modal transfer on image queries for visual question answering tasks derived from VQAv2.*
> > > |Model|Food-Class|Shirt-Color|Man-Holding|Avg.|
> > > |-|-|-|-|-|
> > > |No Context|0.00|0.00|0.00|0.00|
> > > |Image ICL Prompt|0.70|0.41|0.46|0.52|
> > > |Image ICL Patch|0.49|0.19|0.39|0.36|
> > > |Text ICL Prompt|0.85|0.48|0.56|0.63|
> > > |**Text ICL Patch**|**0.93**|**0.56**|**0.59**|**0.69**|
> > >
> > > [1] Goyal et. al. Making the V in VQA Matter: Elevating the Role of Image Understanding in Visual Question Answering. CVPR 2017.\
> > > [2] Marino et. al. OK-VQA: A Visual Question Answering Benchmark Requiring External Knowledge. CVPR 2019. \
> > > [3] Schwenk et. al. A-OKVQA: A Benchmark for Visual Question Answering using World Knowledge. ECCV 2022.

---

### Official Review · Reviewer_2rbY · 2025-03-13

**Overall Recommendation:** 3

**Summary:**

The paper studies the capability of task representation sharing/transfer between VLM and LLM. The authors identify a 'task vector' (the delimiter token between the last query-answer pair) in one modality, transfer it to the other modality, and test the model capacity to achieve the given task without additional prompting or fine tuning. Six cross-modal tasks are constructed and several cross modal evaluation procedures are evaluated (eg, LLM -> VLM).  Three multi-modal architectures are considered (LLava, ideafics and Mantis-Fuyu). Quantitative evaluation is reported using 100 test samples.

## update after rebuttal
Thanks to the authors for the rebuttal, including clarifications and additional experiments. Those additional studies consolidate the paper.

**Claims And Evidence:**

The authors aim at demonstrating that there exists a "shared task vector, which is invariant to modality (text, image) and format (examples, instructions)"
Experiments are designed to demonstrate this claim.
My comments are:
1) VLM are precisely learnt so that the image modality (via the image encoder) and the text modality (via the text encoder) are aligned in a shared vector space. Consequently, the claims and the conclusions are not surprising
2) the claim of the existence of a 'task-vector' (which the authors borrow from previous work), is in my view problematic: if the exact position of the task vector is not consistent across tasks and modalities, then i doubt that we can call it a 'task vector;

**Essential References Not Discussed:**

not to my knowledge.

**Ethical Review Concerns:**

no ethical concern

**Experimental Designs Or Analyses:**

There is a certain level of rigor in the experimental design, and an attempt of fair comparisons. The different setups are quite thorough, and  the experimental procedure seems well thought in general, thought their precise description  is not always clear to me.
 I expect the authors to publish their code, so that the experiments can be reproduced.
Ablation study is reported in the annex.

**Methods And Evaluation Criteria:**

Several evaluation procedure are proposed. Quantitative evaluation is given by accuracy, a reasonable and classical metric.

**Other Comments Or Suggestions:**

The evaluation protocoles and results are interesting, however, the paper would gain in 1) showing that their conclusion is general, and not limited to the several tasks (which are limited in scope), 2) clarifying the description, re-writing part of the paper (experiments section).

**Other Strengths And Weaknesses:**

The paper's idea and direction are definitively very interesting. It attempts to analyse the underlying behavior of the model (ie transparency analysis).  It is however a very 'descriptive' paper, which makes it at times hard to follow.  I am not fully convinced of the conclusions that are drawn.

**Questions For Authors:**

Please could you clarify how the  "token representation curves" (figure 14 and 15) are generated.

**Relation To Broader Scientific Literature:**

Little work has been done regarding the cross-modality transfer. The authors cite relevant papers.

**Theoretical Claims:**

The paper is experimental, trying to analyse the behavior of the models.

---

> ### Author Rebuttal · Authors · 2025-04-01
>
> Thank you for your thoughtful feedback. Below, we report additional results that address your concerns, including evaluation on in-the-wild tasks, and clarifications of the experimental description.
>
> ---
>
> > 1. [T]he claims and the conclusions are not surprising, as it is straightforward that VLM maps inputs from different modalities into a shared space.
>
> We study VLMs trained to map image embeddings into the representation space of an LLM.
> - Although this can be simplified as mapping different modalities into a shared space, this perspective does not explain our results.
> - **In fact, Figure 2a of the main text shows that the image and text embeddings don’t cluster by task. In contrast, Figure 2b shows that the task vector at the end of the sequence does group by task.**
> - This grouping in Figure 2b is not supervised in any way – its emergence is driven only by the next token prediction loss, or the fact that the answer is conceptually similar.
> - We find it striking that these task representations, learned without explicit supervision, are aligned across modalities (image, text) and specifications (examples, instructions). We believe these are new and valuable insights that can inform future research.
>
> ---
>
> > 2. [I]f the exact position of the task vector is not consistent across tasks and modalities, then [I] doubt that we can call it a [‘task vector’]
>
> We would like to clarify that **the layer position from which we extract the task vector is consistent across tasks and modalities**, as seen in Table 1 below. We determine this hyperparameter via average accuracy across all tasks, as discussed in L210 of the main text.
>
> *Table 1. Layer position of task vector, by VLM.*
> | LLaVA-v1.5 | Mantis-Fuyu | Idefics2 |
> |-|-|-|
> |15|23|16|
>
> ---
>
> > 3. I expect the authors to publish their code
>
> We will indeed publish the code to ensure reproducibility.
>
> ---
>
> > 4. [Show] that their conclusion is general, and not limited to the several tasks (which are limited in scope)
>
> **In Sec. A.4 of the Appendix we include an evaluation based on VQAv2, which represents a more in-the-wild set of tasks.** We copy the results to Table 2, showing that patching is also effective in more general VQA tasks. Following this discussion, we will move this result to the main text. [We also invite you to view Table 2 in our response to Reviewer CQRE](https://openreview.net/forum?id=77ziPGdQct&noteId=fxPChMsZmQ), which includes experiments on four new tasks.
>
> *Table 2. We show the test accuracy of cross-modal transfer on image queries for visual question answering tasks derived from VQAv2.*
> |Model|Food-Class|Shirt-Color|Man-Holding|Avg.|
> |-|-|-|-|-|
> |No Context|0.00|0.00|0.00|0.00|
> |Image ICL Prompt|0.70|0.41|0.46|0.52|
> |Image ICL Patch|0.49|0.19|0.39|0.36|
> |Text ICL Prompt|0.85|0.48|0.56|0.63|
> |Text ICL Patch|**0.93**|**0.56**|**0.59**|**0.69**|
>
> ---
>
> > 5. [Clarify] the description, [re-write] part of the paper (experiments section).
>
> Thank you for your comments; we will revise our experiments to improve their clarity.
>
> Additionally, we will include the pseudocode used to produce Figures 8-9,14-15 to the paper (further described below), and include a link to our code, so that the precise experimental procedure is available to readers.
>
> ---
>
> > 6. Please could you clarify how the "token representation curves" (figure 14 and 15) are generated.
>
> To produce the token representation curves, we do the following. For each task, we cache the layer activations and project them with the model’s existing final normalization layer and unembedding matrix [1], which produces a probability distribution over the entire vocabulary. We isolate the scores for three pre-defined tokens – input, task, and answer – and apply a softmax to produce relative probabilities. We then plot the mean and variance of these relative probabilities at each layer.
>
> **We also provide the pseudocode below.**
> ```
> def continuous_rep_evolution(model, dataset, select_vocab_idx):
>     """
>     Plots the relative probability of the input, task, and answer token (given by
>     `select_vocab_idx`) across layers of `model`, for a given `dataset` representing a task.
>     """
>     dataset_rel_prob = []
>     for sample in dataset:
>         feats = cache_act(model(sample)) # dim is [num_layers, 1, hidden_dim]
>         feats = model.norm(feats)
>         token_dist = model.lm_head(feats) # dim is [num_layers, 1, vocab_size]
>         input_idx, task_idx, answer_idx = select_vocab_idx
>         token_dist = token_dist[:, :, [input_idx, task_idx, answer_idx]]  # dim is [num_layers, 1, 3]
>         rel_prob = softmax(token_dist)
>         dataset_rel_prob.append(rel_prob)
>     plot_layer_vs_rel_prob(dataset_rel_prob)
> ```
>
> ---
>
> References\
> [1] nostalgebraist. interpreting gpt: the logit lens. LessWrong 2020.

---

### Official Review · Reviewer_CQRE · 2025-03-14

**Overall Recommendation:** 3

**Summary:**

This paper explores how autoregressive vision-language models (VLMs) form cross-modal task representations, which it identifies as "task vectors." These vectors efficiently encode task information across text and image inputs, enabling effective cross-modal transfer. The authors demonstrate task vectors surpass traditional few-shot prompting, transfer seamlessly from language-only models to vision-language models, and show enhanced efficiency when combining textual instructions with examples.

**Claims And Evidence:**

The claims in this paper are convincingly supported by rigorous experiments and empirical analysis. Results consistently demonstrate that task vectors outperform standard few-shot prompting across multiple tasks and models, clearly validating the core claims about cross-modal transfer capabilities.

**Essential References Not Discussed:**

The paper adequately cites relevant literatures.

**Experimental Designs Or Analyses:**

Experimental designs are robust and systematically presented. The quantitative evaluation of different modalities and methods (instruction-based, example-based, and ensemble patching) effectively demonstrates the superiority and generalizability of the proposed cross-modal task vectors. However, broader task variety would be beneficial to strengthen claims further.

**Methods And Evaluation Criteria:**

The proposed method of cross-modal task vector patching and the selection of evaluation tasks are well-suited for examining VLM capabilities. The benchmark tasks are diverse enough to illustrate meaningful differences and strengths of cross-modal representations, providing appropriate evaluation criteria.

**Other Comments Or Suggestions:**

No I don't have any other comments or suggestions.

**Other Strengths And Weaknesses:**

## Strengths
- Clearly demonstrates cross-modal alignment through systematic and compelling experimental validation.
- The method of task vector extraction and patching is innovative and well-explored empirically.
- Enhances interpretability and understanding of VLMs significantly through clear presentation and insightful analyses.

## Weaknesses
- **Limited Task Complexity**: While the paper claims robustness in task vectors, the experimental setup relies predominantly on simplified tasks (e.g., mapping capitals to countries or matching foods with colors). It remains unclear if these findings generalize well to more complex multimodal tasks that involve nuanced reasoning or domain-specific knowledge. Simple tasks may not reflect real-world challenges. I'd recommend adding more complex tasks to the evaluation like such as VQAv2.
- **Task Overriding**: The "task overriding" experiment in Section 4.4 presents compelling qualitative examples (Figure 7) showing that patching can supersede an original task in the prompt, but the quantitative backing is insufficient. Table 4 reports results on only 100 random pairs of conflicting questions from VQAv2, with a single accuracy metric (0.32 for Instruction Patch vs. 0.05 for System Prompt). This limited evaluation does not demonstrate the general effectiveness of task overriding across diverse scenarios, such as tasks with varying degrees of conflict (e.g., semantic vs. syntactic conflicts) or different task domains (e.g., factual recall vs. creative generation). For instance, the paper does not test whether patching can override tasks in cases where the original task is deeply ingrained in the model’s pre-training (e.g., answering "What is the weather like?" when overridden to "What is the historical significance of this location?"). To address this, the authors should expand the quantitative evaluation to include a broader range of conflicting task pairs, stratified by conflict type and domain, and report additional metrics, such as the proportion of outputs that partially retain the original task, to assess the robustness of overriding.

- **Missing Practical Considerations**: The paper does not discuss practical considerations for deploying cross-modal task vectors, such as computational overhead of patching, robustness to noisy inputs, or sensitivity to instruction phrasing. These are critical for real-world applicability, especially given the Impact Statement’s emphasis on accessibility.

**Questions For Authors:**

- Have you explored or considered the robustness of task vectors under substantial domain shifts, such as specialized domains like medical or scientific imagery?
- Could you provide further quantitative or analytical insights into the potential reasons behind the observed representational convergence across modalities, especially regarding model architecture and training procedures?

**Relation To Broader Scientific Literature:**

The contributions of this paper build meaningfully on existing literature regarding mechanistic interpretability, particularly in-context learning and interpretability in vision-language contexts. The discussion effectively situates the contributions within current frameworks and models, such as those by [1] and [2], without redundancy.

[1] Hendel, R., Geva, M., and Globerson, A. In-context learning creates task vectors. Findings of Empirical Methods in Natural Language Processing, 2023.

[2] Todd, E., Li, M. L., Sharma, A. S., Mueller, A., Wallace, B. C., and Bau, D. Function vectors in large language models. International Conference on Learning Representations, 2024.

**Theoretical Claims:**

The paper does not present explicit theoretical proofs. Therefore, theoretical claims are not a concern in this context.

---

> ### Author Rebuttal · Authors · 2025-04-01
>
> Thank you for your constructive feedback. Following your suggestions, we’ve added **four new experiments** on task complexity, task overriding, and practical considerations.
>
> ---
>
> > 1. [Add] more complex tasks […] such as VQAv2.
>
> **In Sec. A.4 of the Appendix we include an evaluation based on VQAv2.** We copy these results to Table 1 below. Table 1 shows that patching is also effective in more complex tasks. Following this discussion, we will move this result to the main text.
>
> *Table 1. Test accuracy of cross-modal transfer on tasks derived from VQAv2.*
> |Model|Food-Class|Shirt-Color|Man-Holding|Avg.|
> |-|-|-|-|-|
> |No Context|0.00|0.00|0.00|0.00|
> |Image ICL Prompt|0.70|0.41|0.46|0.52|
> |Image ICL Patch|0.49|0.19|0.39|0.36|
> |Text ICL Prompt|0.85|0.48|0.56|0.63|
> |Text ICL Patch|**0.93**|**0.56**|**0.59**|**0.69**|
>
> ---
>
> > 2. [D]emonstrate [...] semantic vs. syntactic conflicts [...] factual recall vs. creative generation
>
> **Table 2 further stratifies task overriding performance, and scales up the number of evaluation samples to 1000.** Patching outperforms system prompting by 27-59\% for conflicts in Semantics, Creative Generation, Factual Recall. For conflicts in Syntax, system prompting significantly improves, likely because syntactic instructions are often not mutually exclusive.
>
> *Table 2. Task conflict stratified by degree of conflict and task domain.*
> |Method|(a) Semantics|(b) Syntax|(c) Creative Generation|(d) Factual Recall|
> |-|-|-|-|-|
> |Original Task|0.10|0.15|0.08|0.15|
> |+ System Prompt|0.09|0.49|0.06|0.15|
> |+ Instruction Patch|**0.36**|**0.59**|**0.65**|**0.42**|
>
> Since some settings are open-ended, we use GPT4o to rate correctness.
> - (a) Same as Table 4 in the main text, scaled to 1000 images.
> - (b) Formatting instructions (e.g., answer in ALL CAPS, quotes, JSON) on same 1000 images.
> - (c) Creative prompts (e.g., invent a book title, character name, company name) on same 1000 images.
> - (d) “Outside knowledge” questions on 148 overlapping images from OKVQA and A-OKVQA.
>
> ---
>
> > 3. [T]est [...] cases where the original task is deeply ingrained in the model’s pre-training.
>
> **Table 3 shows that patching can still override deeply ingrained tasks.**
>
> *Table 3. Stratification of 1000 examples from Table 2a by level of ingrainment.*
> |Method|Highly Ingrained|Moderately Ingrained|Lightly Ingrained|
> |-|-|-|-|
> |Original Task|0.10|0.11|0.04|
> |+ System Prompt|0.04|0.09|0.08|
> |+ Instruction Patch|**0.20**|**0.36**|**0.38**|
>
> We measure ingrainment by the question perplexity (PPL), and denote tasks with bottom 5% PPL as Highly Ingrained, middle 90% as Moderately Ingrained, and top 5% as Lightly Ingrained.
>
> ---
>
> > 4. [Discuss] computational overhead of patching.
>
> **Table 4 shows the computational overhead of patching.** Patching cuts runtime by 11x and VRAM by 2.4x compared with few-shot prompting, for long text descriptions (see Sec. A.4 of the main text). After injecting the single task vector, the VLM no longer needs to attend to the long context. While computing the task vector requires an upfront cost, it is amortized in future runs.
>
> *Table 4. Overhead of a single forward pass on N=30 Text ICL examples, averaged over 100 runs.*
> |Method|Runtime (seconds)|VRAM (GB)|
> |-|-|-|
> |Prompting (Context + Query)|2.20|20.02|
> |Patching (Task Vector + Query)|0.20|8.21|
> |Query Only|0.19|8.21|
>
> ---
>
> > 5. [Discuss] robustness to noisy inputs, or sensitivity to instruction phrasing.
>
> **Table 5 shows the robustness of patching to noisy instructions.** As one would expect, the performance degrades as the number of typos increases. However, even with typos, patching maintains non-negligible performance.
>
> *Table 5. Accuracy of patching for instructions with varying levels of typos.*
> |Num Character Swaps|Country-Capital|Country-Currency|Animal-Latin|Animal-Young|Food-Color|Food-Flavor|Avg|
> |-|-|-|-|-|-|-|-|
> |s=0|0.58|0.22|0.34|0.44|0.48|0.29|**0.39**|
> |s=1|0.65|0.07|0.33|0.51|0.52|0.13|**0.37**|
> |s=2|0.63|0.14|0.36|0.41|0.48|0.08|**0.35**|
>
> ---
>
> > 6. Have you explored [...] robustness [...] under substantial domain shifts
>
> We have not yet explored domain shifts, but we agree it is a compelling future direction.
>
> ---
>
> > 7. Could you provide [...] potential reasons behind [...] representational convergence
>
> One possibility is that multi-task learning drives compression. VLMs, trained via next-token prediction on diverse web data, implicitly learn multiple tasks at once [1]. Since the same task can be defined in many different ways, and memorizing every variation is impractical, some form of representation sharing is needed to manage this complexity. Other studies have also arrived at this hypothesis [2].
>
> We are also interested in a deeper analysis, but this would require a dedicated study of its own, so we leave it to future work as noted in Sec. 6 of the main text.
>
> ---
>
> References\
> [1] Brown et. al. Language Models are Few-Shot Learners. NeurIPS 2020.\
> [2] Huh et. al. The Platonic Representation Hypothesis. ICML 2024.

---

> > ### Comment · Reviewer_CQRE · 2025-04-08
> >
> > Thank you for your thorough and detailed rebuttal, and for addressing my concerns with careful consideration and meaningful additional analyses.
> >
> > Regarding the evaluation of task complexity based on the VQAv2 dataset, I appreciate your clarification and the decision to highlight these additional evaluations by moving them from the Appendix to the rebuttal. However, to clearly restate my original point, my suggestion was aimed at including evaluations or metrics that could capture a **broader spectrum of complexity** present in the full VQAv2 dataset. While the selected tasks `("Food-Class," "Shirt-Color," "Man-Holding")` derived from VQAv2 indeed introduce complexity beyond the original simpler tasks, they predominantly focus on *single-attribute recognition* or relatively straightforward *object-level identification*. Incorporating additional evaluation tasks requiring *multi-step reasoning*, *subtle semantic distinctions*, or *interactions among multiple objects* would further substantiate your claims about generalizability across the full complexity spectrum of **realistic VQA scenarios**.
> >
> > Nevertheless, the additional experiments and analyses you provided, particularly regarding task overriding stratification and practical considerations such as **computational efficiency** and **robustness to noisy instructions**, have considerably strengthened the author's manuscript. Given these meaningful improvements and your responsiveness, I positively revise my original assessment from `"Weak Reject"` to `"Weak Accept"`. I also encourage explicitly highlighting future directions that address robustness under **substantial domain shifts** or more nuanced multimodal reasoning tasks.

---

### Official Review · Reviewer_fXQS · 2025-03-14

**Overall Recommendation:** 4

**Summary:**

The paper explores how VLMs create cross-modal task representations that are invariant to input modality (text or image) and format (examples or instructions). These task vectors, derived from one modality, can effectively trigger task execution in another. It often outperforms traditional few-shot prompting. The study also shows that task vectors can transfer from base language models (LLMs) to fine-tuned VLMs and can be defined using instructions alone. These findings reveal that VLMs map diverse inputs into shared semantic representations, enhancing their flexibility and efficiency.

**Claims And Evidence:**

The paper presents evidence supporting its claims about cross-modal task representations in VLMs. It demonstrates that VLMs create shared task vectors invariant to input modality and format, which outperform traditional few-shot prompting. The authors show that these task vectors can transfer from base LLMs to fine-tuned VLMs and can be derived from instructions alone. Experiments, including cross-modal patching and task overriding, provide robust evidence for these claims. The findings reveal how VLMs map diverse inputs into common semantic representations, enhancing their flexibility and efficiency.

**Essential References Not Discussed:**

This paper provides a good review of related studies. However, there are several essential related works that are not currently cited or discussed in the paper, which could provide additional context and depth to the key contributions. Here are some examples:

1. Lin Z, Yu S, Kuang Z, et al. Multimodality helps unimodality: Cross-modal few-shot learning with multimodal models[C]//Proceedings of the IEEE/CVF Conference on Computer Vision and Pattern Recognition. 2023: 19325-19337.

2. Doveh S, Perek S, Mirza M J, et al. Towards multimodal in-context learning for vision & language models[J]. arXiv preprint arXiv:2403.12736, 2024.

**Experimental Designs Or Analyses:**

The experimental designs and analyses are methodologically sound and provide a few insights (see Findings) into the cross-modal alignment of task representations in VLMs. The use of multiple VLM architectures and a diverse set of tasks ensures that the findings are not model-specific and can be generalized.

However, there are areas for improvement: increasing the sample sizes for validation and testing would enhance the statistical power and reliability of the results. Incorporating more complex, multi-step reasoning tasks could offer deeper insights into the models' capabilities. Evaluating a broader range of VLM and LLM pairs would strengthen the conclusions regarding the preservation of task representations during fine-tuning. Additionally, supplementing the visualizations with more quantitative measures of representation alignment would provide a more rigorous assessment of how different modalities are mapped into shared semantic spaces. Addressing these aspects would further solidify the findings and enhance their applicability to real-world scenarios.

**Methods And Evaluation Criteria:**

The methods and evaluation criteria used in this paper are aligned with the research goals and provide robust evidence for the claims made. The use of cross-modal patching, evaluation of instruction-based task vectors, and transfer from LLMs to VLMs, combined with appropriate metrics like accuracy and cosine similarity, make this study comprehensive and insightful. The findings are good contributions to the understanding of how VLMs process and align task representations across modalities, and the proposed methods are likely to inspire further research in this area.

**Other Comments Or Suggestions:**

N/A

**Other Strengths And Weaknesses:**

Pro:
1. The paper is generally well-written. I really appreciate the step-by-step experimental and analytical approach taken in this paper.
2. The paper evaluates a diverse set of VLMs, including both early-fusion and late-fusion models (LLaVA-v1.5, Mantis-Fuyu, Idefics2). This comprehensive evaluation ensures that the findings are not specific to a single model architecture and enhances the generalizability of the results.
3. The authors design a set of six cross-modal tasks that cover a range of semantic relationships. This diversity in tasks helps in understanding the robustness of the proposed methods across different types of tasks.
4. The findings have practical implications for improving the efficiency and flexibility of VLMs. For example, the ability to derive task vectors from instructions alone can significantly reduce the need for large datasets, making the models more accessible and easier to deploy in real-world applications.
5. The demonstration that task vectors can be transferred from LLMs to VLMs suggests that pre-trained language models can be effectively leveraged to enhance multimodal tasks, which is a promising direction for future research and development.

Con:
1. The sample sizes for validation and testing are relatively small (30 for validation and 100 for testing). This limits the statistical power and generalizability of the results. Larger sample sizes would provide more reliable estimates of model performance and enhance the robustness of the findings.
2. The tasks used in the experiments are relatively simple and may not fully capture the complexity of real-world applications. More complex, multi-step reasoning tasks could provide deeper insights into the models' capabilities and limitations.
3. The quality and clarity of instructions used to derive task vectors are critical. The paper assumes that instructions are well-formed and unambiguous, which may not always be the case in real-world scenarios. Ensuring high-quality, unambiguous instructions is essential for the validity of instruction-based task vectors.
4. The study does not explore the impact of different instruction formats or the optimal balance between instructions and examples. This could be an important area for future research to improve the effectiveness of instruction-based learning.

**Questions For Authors:**

See "Other Strengths And Weaknesses"

**Relation To Broader Scientific Literature:**

The paper advances the understanding of how VLMs process and align task representations across different modalities. It builds on prior work in vision-language alignment, mechanistic interpretability, and in-context learning by introducing the concept of cross-modal task vectors. These vectors, derived from either text or image inputs, are shown to be invariant to modality and can be effectively transferred between language models and VLMs. The study extends activation patching techniques to a cross-modal context, demonstrating that task vectors can induce correct task-specific outputs even when applied to a different modality. Additionally, the paper shows that task vectors can be efficiently derived from instructions, offering a more sample-efficient alternative to example-based learning.

**Theoretical Claims:**

N/A

---

> ### Author Rebuttal · Authors · 2025-04-01
>
> Thank you for your thorough and constructive feedback. Below, we add results on larger sample sizes, real-world tasks, and malformed instructions, following your suggestions.
>
> ---
>
> > 1. [T]here are several essential related works that are not currently cited
>
> We will be sure to cite and discuss these related works in the final manuscript.
>
> ---
>
> > 2. The sample sizes for validation and testing are relatively small
>
> **We repeat the task overriding evaluation (Table 4 of the main text) with a larger sample size, increasing it from 100 to 1000 examples in Table 2 below.**
>
> However, for our evaluation tasks in Table 1 of the main text, the sample sizes are unfortunately upper-bounded by real-world constraints, especially our ability to manually cross-check the input-output pairings with online sources as described in Sec. A.1 of the Appendix.
>
> *Table 2. Task overriding results on VQAv2, scaled from 100 to 1000 examples.*
> |          Method                       | Accuracy |
> |---------------------------------|------------------------|
> | Original Task                   | 0.10                   |
> | Original Task + System Prompt   | 0.09      |
> | Original Task + Instruction Patch | **0.36**      |
>
> ---
> > 3. The tasks used in the experiments are relatively simple and may not fully capture the complexity of real-world applications.
>
> **In Sec. A.4 of the Appendix we include an evaluation based on VQAv2, which represents a more in-the-wild set of tasks.** We copy Table 10 from the Appendix to Table 1 below for your convenience. In this experiment, we derive the tasks from questions from VQAv2, where each ICL example is composed of either complex real-world images or dense text descriptions as input. Consistent with Table 2 in the main text, Table 1 below shows that cross modal patching outperforms few-shot prompting.
>
> We agree that multimodal multi-step reasoning tasks, which likely require strong language capabilities when analyzing images, could benefit from our method and are an interesting avenue for future work.
>
> *Table 1. We show the test accuracy of cross-modal transfer on image queries for visual question answering tasks derived from VQAv2.*
> | Model         | Food-Class | Shirt-Color | Man-Holding | Avg. |
> |--------------------|----------------|-----------------|-----------------|----------|
> | No Context       | 0.00           | 0.00           | 0.00           | 0.00     |
> | Image ICL Prompt   | 0.70           | 0.41           | 0.46           | 0.52     |
> | Image ICL Patch    | 0.49           | 0.19           | 0.39           | 0.36     |
> | Text ICL Prompt   | 0.85           | 0.48           | 0.56           | 0.63     |
> | Text ICL Patch | **0.93**       | **0.56**       | **0.59**       | **0.69** |
>
> ---
>
> > 4. The paper assumes that instructions are well-formed and unambiguous
>
> **In Table 5, we examine cross-modal patching’s robustness to malformed and ambiguous instructions by adding typos at varying levels to the instruction.** As one would expect, the performance degrades as the number of typos increases. However, even with typos, patching is able to maintain non-negligible performance.
>
> *Table 5. Robustness to noisy instructions. We randomly swap consecutive word characters in the instruction, following the protocol of [1]. We report the accuracy of cross-modal patching onto image queries with these noisy instructions.*
> | Num Character Swaps       | Country-Capital | Country-Currency | Animal-Latin | Animal-Young | Food-Color | Food-Flavor | Avg  |
> |-----------------------|------------------|-------------------|--------------|--------------|-------------|--------------|------|
> | s=0      | 0.58  | 0.22  | 0.34  | 0.44  | 0.48  | 0.29  | **0.39**  |
> | s=1      | 0.65  | 0.07  | 0.33  | 0.51  | 0.52  | 0.13  | **0.37**  |
> | s=2      | 0.63  | 0.14  | 0.36  | 0.41  | 0.48  | 0.08  | **0.35**  |
>
> ---
>
> > 5. The study does not explore the impact of different instruction formats or the optimal balance between instructions and examples. This could be an important area for future research
>
> We agree that it would be worthwhile to study a larger set of instruction formats, such as ablating whether the instruction is stated as a question versus command or the instruction length.
>
> While we conducted preliminary investigation of the balance between instructions and the number of examples in Figure 6 of the main text, we agree that further exploration – such as testing different weightings when averaging the two vectors – would be interesting.
>
> We would like to thank the reviewer for the exciting and promising suggestions for future research.
>
> ---
>
> References\
> [1] https://github.com/ranvijaykumar/typo

---

> > ### Comment · Reviewer_fXQS · 2025-04-02
> >
> > I sincerely appreciate the authors' responses. Most of my concerns have been addressed. I also hope the authors can incorporate the related studies into the paper and discuss them. Considering that this work has some innovation and provides appropriate analysis, I will maintain my original score.

---

### Decision · Program_Chairs · 2025-05-01

**Decision:**

Accept (poster)

**Comment:**

This paper presents a detailed analysis of how Vision-Language Models (VLMs) generate cross-modal task representations that are invariant to modality and format, which significantly enhances the efficiency of these models. The authors introduce the concept of "task vectors" that can be transferred across modalities, demonstrating that these vectors outperform traditional few-shot prompting. They also provide strong experimental evidence supporting their claims and show the potential for task vectors to be derived solely from instructions, without requiring examples.

Reviewers generally agree on the novelty of the approach, praising the clarity and comprehensiveness of the experiments.  Overall, the paper makes valuable contributions to the field of VLMs and offers promising directions for future research. The results are well-supported by robust experiments, and the findings are likely to inspire further work in multimodal learning and task vector representations.

The AC agrees with the reviewers to accept the submission.